# GEOMETRIC PROPERTIES OF NEURAL MULTIVARIATE REGRESSION: AN EMPIRICAL STUDY

**George Andriopoulos**[1*]    **Zixuan Dong**[2,3*†]    **Bimarsha Adhikari**[1*]    **Keith Ross**[1]

[1] New York University Abu Dhabi
[2] SFSC of AI and DL, NYU Shanghai
[3] New York University

## ABSTRACT

Neural multivariate regression underpins a wide range of domains such as control, robotics, and finance, yet the geometry of its learned representations remains poorly characterized. While neural collapse has been shown to benefit generalization in classification, we find that analogous collapse in regression consistently degrades performance. To explain this contrast, we analyze models through the lens of intrinsic dimension. Across control tasks and synthetic datasets, we estimate the intrinsic dimension of last-layer features ($ID_H$) and compare it with that of the regression targets ($ID_Y$). Collapsed models exhibit $ID_H < ID_Y$, leading to over-compression and poor generalization, whereas non-collapsed models typically maintain $ID_H > ID_Y$. For the non-collapsed models, performance with respect to $ID_H$ depends on the data quantity and noise levels. From these observations, we identify two regimes—over-compressed and under-compressed—that determine when expanding or reducing feature dimensionality improves performance. Our results provide new geometric insights into neural regression and suggest practical strategies for enhancing generalization.

## 1 INTRODUCTION

Neural multivariate regression has emerged as a cornerstone of modern machine learning, powering a wide spectrum of applications, including imitation learning, robotic control, financial prediction, and reinforcement learning value approximation. In this work, we empirically investigate the *geometric structure of neural multivariate regression*, with an emphasis on the geometry of last-layer feature vectors.

Prior efforts have largely framed this problem through the lens of *neural collapse*. In classification, Neural Collapse (NC) describes the emergence of a highly symmetric configuration: last-layer features converge to the vertices of a Simplex Equiangular Tight Frame (ETF), aligned with the classifier weights (Papyan et al., 2020). In regression, by contrast, Neural Regression Collapse (NRC) manifests

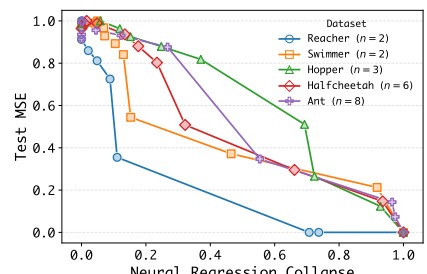

Figure 1: Neural Regression Collapse correlates with high Test MSE. The smaller the NRC value, the closer features lie to the $n$-dimensional subspace.

as the concentration of last-layer features within a linear subspace spanned by the top $n$ principal components of the last-layer feature matrix, where $n$ is the number of target variates. Since $n$ is typically much smaller than the feature dimension, regression collapse implies a major reduction in representational degrees of freedom (Andriopoulos et al., 2024).

In this paper, we first make a key empirical observation: *Collapse, beneficial in classification[1], harms regression models to consistently exhibit degraded generalization as compared to their non-collapsed*

---

*Equal contribution. †Corresponding author: zixuandong@nyu.edu
[1]Appendix B carefully review related work and discuss generalization of neural collapse for classification.

*counterparts.* Figure 1 illustrates this, showing high values of test MSE for models with highly collapsed features (low values of NRC metric) for five robotic locomotion tasks. Existing theoretical and empirical treatments of regression collapse, including the work of Andriopoulos et al. (2024), do not account for this degradation. This raises a central open question: ***Why does neural collapse hinder generalization in multivariate regression, in contrast to its beneficial role in classification?***

We address this question by employing *intrinsic dimension (ID)*, which quantifies the effective dimensionality of the manifold in which the data lies. As shown in Figure 2 and studied in the paper, intrinsic dimension can capture nonlinearities that the PCA approach of NRC cannot, and thus reveals a more refined geometry of multivariate regression. While intrinsic dimension has been previously studied in the context of neural classification (Ansuini et al., 2019), to the best of our knowledge, this is the first work to analyze neural multivariate regression from this perspective.

## 2 BACKGROUND AND KEY METRICS

We consider the multivariate regression problem with $M$ training examples $\{(\mathbf{x}_i, \mathbf{y}_i), i = 1, ..., M\}$, where each input $\mathbf{x}_i$ belongs to $\mathbb{R}^D$ and each target vector $\mathbf{y}_i$ belongs to $\mathbb{R}^n$. Considering a deep regression network of the form:

$$f_{\theta,\mathbf{W},\mathbf{b}}(\mathbf{x}) = \mathbf{W}\mathbf{h}_\theta(\mathbf{x}) + \mathbf{b},$$

where $\mathbf{h}_\theta(\cdot) : \mathbb{R}^D \to \mathbb{R}^d$ is the non-linear multi-layer feature extractor, $\mathbf{W} \in \mathbb{R}^{n \times d}$ represents the final linear layer in the model, and $\mathbf{b} \in \mathbb{R}^n$ is the bias vector. The parameters $\theta, \mathbf{W}, \mathbf{b}$ are all trainable. We typically train the DNN using gradient descent to minimize the regularized L2 loss:

$$\min_{\theta,\mathbf{W},\mathbf{b}} \frac{1}{2M} \sum_{i=1}^{M} ||f_{\theta,\mathbf{W},\mathbf{b}}(\mathbf{x}_i) - \mathbf{y}_i||_2^2 + \frac{\lambda_{WD}}{2}(||\theta||_2^2 + ||\mathbf{W}||_F^2),$$

where $|| \cdot ||_2$ and $|| \cdot ||_F$ denote the $L_2$-norm and the Frobenius norm, respectively; and $\lambda_{WD}$ denotes weight decay parameter.

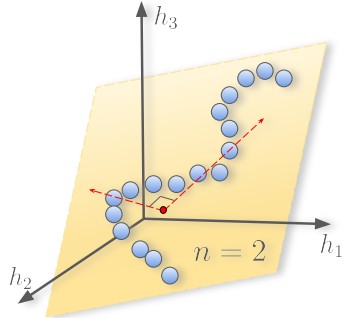

Figure 2: When the target dimension is $n = 2$, the collapsed features (blue points) with intrinsic dimension of 1 can lie close to a subspace (yellow plane) spanned by the first 2 principal components (red arrows) of the features.

**Neural Regression Collapse: NRC1 metric**   Let $\mathbf{h}_i := \mathbf{h}_\theta(\mathbf{x}_i)$ be the feature vector associated with example $\mathbf{x}_i$, $i = 1, \ldots, M$. Further let $\widetilde{\mathbf{h}}_i$ be the centered normalized feature vector, that is, $\widetilde{\mathbf{h}}_i := (\mathbf{h}_i - \bar{\mathbf{h}}) \cdot ||\mathbf{h}_i - \bar{\mathbf{h}}||^{-1}$ where $\bar{\mathbf{h}} := M^{-1} \sum_{i=1}^{M} \mathbf{h}_i$. For any $p \times q$ matrix $\mathbf{C}$ and any $p$-dimensional vector $\mathbf{v}$, let $\text{proj}(\mathbf{v}|\mathbf{C})$ denote the projection of $\mathbf{v}$ onto the subspace spanned by the columns of $\mathbf{C}$. Let $\mathbf{H}_{\text{PCA}}$ be the $d \times n$ matrix with the columns consisting of the first $n$ PCs of the feature matrix $\mathbf{H}$. The NRC1 metric is defined as

$$\text{NRC1} := \frac{1}{M} \sum_{i=1}^{M} ||\widetilde{\mathbf{h}}_i - \text{proj}(\widetilde{\mathbf{h}}_i|\mathbf{H}_{\text{PCA}})||_2^2,$$

which measures the extent to which the last-layer features concentrate around their top $n$ principal components. A model is considered collapsed if NRC1 is small, indicating that the features lie almost entirely within an $n$-dimensional subspace. Non-collapsed models have higher values of NRC1, differing from those of collapsed models by orders of magnitude. Andriopoulos et al. (2024) demonstrated that slightly increased weight decay $\lambda_{WD}$ quickly leads to model collapse during training. In Appendix G, we empirically demonstrate how weight decay, dropout regularization, and model depth influence NRC1 and $ID_H$.

**Intrinsic Dimension via 2-NN Estimation**   To uncover the finer geometry of the learned features, beyond what linear methods like PCA reveal, we turn to intrinsic dimension — the minimal number of degrees of freedom needed to describe the data without significant information loss. To estimate the intrinsic dimension, we use the 2-NN estimator, introduced by Facco et al. (2017). For a given point, let $r_1$ and $r_2$ denote the distances to its first and second nearest neighbors; define the ratio $\mu := r_2/r_1$. Under the assumption of locally uniform sampling, the cumulative distribution of the ratio follows a Pareto distribution with parameter $d$: $F(\mu) = 1 - \mu^{-d}$ for $\mu \geq 1$. The intrinsic dimension $d$ is then estimated by linear regression. Details and intuitions are provided in the Appendix C.

## 3 DATASETS

We perform experiments on robotic locomotion and vision-based datasets described in this section. Full experimental details are included in Appendix A. Moreover, Appendix H examines four more challenging tasks with varying sizes, increased intrinsic dimensions, and visual inputs.

**MuJoCo locomotion** MuJoCo (Brockman et al., 2016; Towers et al., 2023) is a widely used physics simulator for continuous-control reinforcement learning. Following Andriopoulos et al. (2024), we use Reacher, Swimmer, and Hopper datasets, and additionally include the higher-dimensional Halfcheetah and Ant datasets from the D4RL benchmark (Fu et al., 2020). Each dataset contains expert demonstrations with proprioceptive state inputs ($\mathbf{x}_i$) and corresponding action targets ($\mathbf{y}_i$). States encode joint positions, angles, velocities, and angular velocities, while actions represent joint torques. We subsample the expert data to form low- and high-data regimes of 1,000 and 20,000 samples, respectively.

**Vision-based regression** We construct two vision-based regression tasks that differ in the amount of task-irrelevant noise present in the targets. In both cases, regression targets are generated by applying a fixed random linear projection to extracted image features. For *MNIST regression*, features are extracted from the penultimate layer of a CNN trained to over 99% accuracy on MNIST, and then projected to 25 dimensions. Because the feature extractor is trained on the same domain, image-feature mismatches are largely suppressed, yielding clean, self-consistent targets. For *CIFAR-10 regression*, features are extracted using a ResNet-18 pretrained on ImageNet and *not* fine-tuned on CIFAR-10, and then projected to 10 dimensions. This domain mismatch causes the projected targets to retain instance-specific, task-irrelevant variation, resulting in noisy targets.

## 4 INTRINSIC DIMENSION AND GENERALIZATION

By NRC1 definition, this metric does not provide insight into whether the features collapse into lower-dimensional non-linear manifolds. To explore this issue, Figure 3 presents scatter plots for the intrinsic dimension of the last-layer features, denoted $ID_H$, versus NRC1 for MuJoCo datasets. The intrinsic dimension of the regression targets, $ID_Y$, is consistently lower than the ambient target dimension $n$. For highly collapsed models with near-zero NRC1, the intrinsic dimension of last-layer features satisfies $ID_H \lesssim ID_Y < n$ and continues to shrink even as NRC1 saturates at zero, indicating that features lie on increasingly lower-dimensional nonlinear manifolds within an $n$-dimensional linear subspace. In contrast, non-collapsed models satisfy $ID_H > ID_Y$, and the intrinsic dimension increases monotonically with NRC1, making NRC1 and intrinsic dimension qualitatively interchangeable. As a result, the following analysis focuses on intrinsic dimension, which reveals a soft threshold at $ID_Y$ that separates two NRC regimes and quantifies the degree of collapse ($ID_H$) across all NRC1 values. Appendix D further illustrates ID evolution during training.

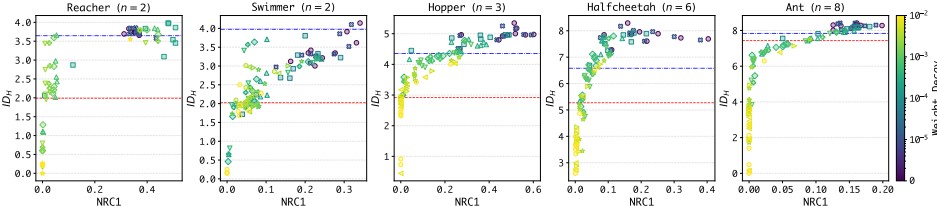

Figure 3: Relationship between NRC1 and intrinsic dimension of the last-layer features $ID_H$. Dots correspond to models trained with different architectures and weight decay parameters, with the colors denoting the degree of weight decay. The horizontal red dashed line is drawn at $ID_Y$, and the blue line is drawn at $ID_X$, the intrinsic dimension of the inputs.

We now turn back to the question raised at the beginning: in Figure 1, why collapsed models suffer increasing generalization error as $ID_H$ (and hence as NRC1) decreases, in contrast to collapse in classification. Figure 4 shows the relationship between $ID_H$ and both training and test MSE for Halfcheetah-1K/20K (1,000 samples/20,000 samples), CIFAR-10, and MNIST datasets. Plots for the remaining datasets are in Appendix F.

**Train MSE decreases when ID$_\mathbf{H}$ increases.** This trend is evident in the first row of Figure 4. To explain, we note that stronger regularization reduces $ID_H$ from Figure 3 and Figure 11. And Theorems 4.1 and 4.3 in Andriopoulos et al. (2024) also tell that stronger regularization reduces the dimension of the linear subspace containing the feature manifold. As a result, reducing $ID_H$ effectively squashes features onto more curved, lower-dimensional manifolds. Because the final linear layer $\mathbf{W}$ can only apply affine transformations, it becomes harder to undo this curvature to match the target manifold, explaining the higher training error.

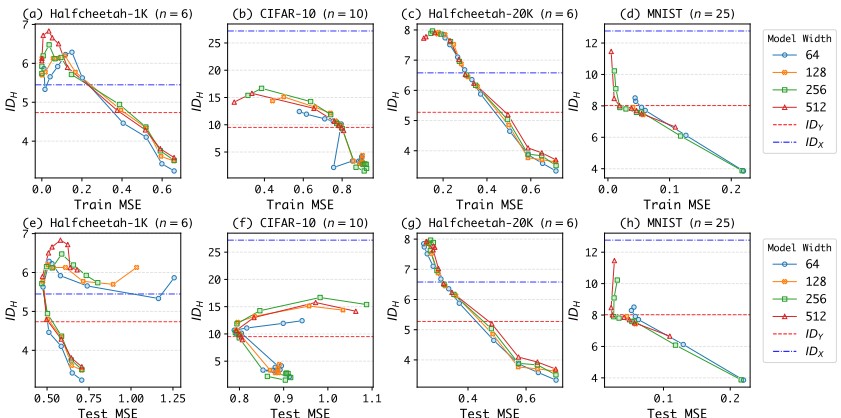

Figure 4: Generalization and Intrinsic Dimension for Halfcheetah, MNIST, and CIFAR-10 datasets.

**Test MSE with respect to ID$_\mathbf{H}$ behaves differently according to its relationship to ID$_\mathbf{Y}$.** The second row of Figure 4 shows fundamental differences between collapsed and non-collapsed models:

- $(ID_H < ID_Y)$: In this regime, the collapsed model's features are confined to a manifold whose intrinsic dimension is lower than that of the targets. This *over-compression* intuitively means the last-layer features lack information essential for reconstructing and generalizing beyond the full target manifold. Theoretical statements in Appendix I show that in this regime, the set of all possible model predictions is a proper subset of the target manifold, thus leading to poor performance on both train and test data for collapsed models.

- $(ID_H \geq ID_Y)$: We distinguish these non-collapsed models between two cases:

  *(i) Low-data or noisy-target tasks.* In this setting (Figs. 4 (e), (f)), the test MSE exhibits a surprising U-shaped dependence on $ID_H$, with a minimum near $ID_H \simeq ID_Y$. When training data are scarce or targets are noisy, $f_\theta$ learns a feature manifold with intrinsic dimension exceeding that of the true feature manifold, driven by the negative effect of outliers. These extra dimensions capture sample-specific noise, leading to overfitting during training. As regularization is reduced—equivalently, as $ID_H$ increases—this overfitting worsens, resulting in higher test MSE.

  *(ii) High-data and low-noise tasks.* In this case, test MSE follows the same trend as train MSE, decreasing monotonically with $ID_H$ (Figs. 4 (g),(h)). To explain, we note that with a large amount of training data and low target noise, $f_\theta$ can fit the training data closely while maintaining smoothness to avoid overfitting, and consequently, the manifold for $\mathbf{H}_{train}$ is similar to the manifold for $\mathbf{H}_{test}$.

## 5 CONCLUSION

In this paper, we take the first step towards a systematic geometric analysis of neural multivariate regression through the lens of intrinsic dimension, highlighting a fundamental contrast with classification. We showed that regression collapse corresponds to an over-compressed regime where the feature manifold has a lower intrinsic dimension than the target manifold, leading to poor generalization. In contrast, non-collapsed models typically satisfy $ID_H \geq ID_Y$, with generalization behavior governed by whether the task is low-data/noisy or high-data/low-noise. These results suggest practical criteria for improving generalization by monitoring and adjusting $ID_H$ in applied multivariate regression[2]. In Appendix B and J, we carefully review related work and discuss limitations and future work.

---

[2]In Appendix K, we present an example of evaluating a behaviorally-cloned policy in robotic simulation and illustrate the relationship between its performance and intrinsic dimension.

ACKNOWLEDGMENTS

This work was supported in part by NYU Abu Dhabi Center for Artificial Intelligence and Robotics and Center for Interdisciplinary Data Science and AI, funded by Tamkeen under the Research Institute Award CG010. It was also partially supported by Shanghai Frontiers Science Center of Artificial Intelligence and Deep Learning at NYU Shanghai.

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

# APPENDIX CONTENTS

# A EXPERIMENT DETAILS

Table 1: Overview of datasets employed in the main body.

| Dataset | Data Size | Input Type | Input Dim ($D$) | Input ID ($ID_X$) | Target Dim ($n$) | Target ID ($ID_Y$) |
|---------|-----------|------------|-----------------|-------------------|------------------|--------------------|
| Swimmer | 20,000 | Raw state | 8 | 3.98 | 2 | 2.00 |
| Reacher | 20,000 | Raw state | 11 | 3.65 | 2 | 1.99 |
| Hopper | 20,000 | Raw state | 11 | 4.35 | 3 | 2.92 |
| Halfcheetah | 20,000 | Raw state | 17 | 6.58 | 6 | 5.27 |
| Ant | 20,000 | Raw state | 111 | 7.83 | 8 | 7.43 |
| MNIST | 50,000 | Grayscale image | $28 \times 28$ | 12.76 | 25 | 8.02 |
| CIFAR-10 | 50,000 | RGB image | $32 \times 32 \times 3$ | 27.20 | 10 | 9.51 |

## A.1 MUJOCO EXPERIMENTS

MuJoCo (Multi-Joint dynamics with Contact) is a physics engine designed for research in robotics, biomechanics, and animation, providing fast and accurate simulations of systems involving complex contact dynamics. It balances physical realism with computational efficiency, enabling reliable modeling of robot–environment interactions (Towers et al., 2024). Environments involved in this work include:

- **Reacher**: A two-jointed robotic arm tasked with moving its tip to a randomly generated target in a 2D plane.
- **Swimmer**: A chain-like robot with three body segments connected by two rotors, aiming to propel itself forward in 2D as quickly as possible.
- **Hopper**: A one-legged, four-part robot that seeks to hop forward at maximum speed in 2D.
- **HalfCheetah**: A planar, bipedal robot with a torso and two legs, each consisting of two joints. It aims to run forward as quickly as possible along a 2D track by coordinating its leg movements.
- **Ant**: A quadrupedal robot with four legs and multiple joints, designed to move in a 3D plane. Its goal is to walk or run forward efficiently, despite the challenge of balancing and coordinating many degrees of freedom. Although Ant's state space has 111 dimensions, 84 of the dimensions related to external contact forces are always zeros in the dataset. Thus, the effective input dimension is 27.

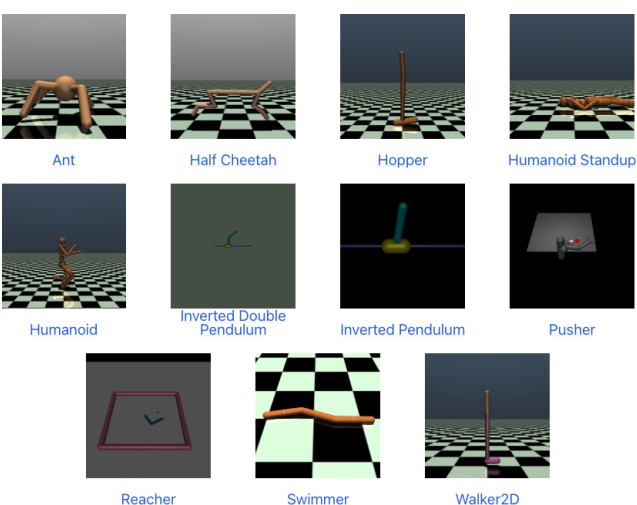

Figure 5: Screenshot of various MuJoCo environments (Towers et al., 2024).

All environments introduce stochasticity by perturbing a fixed initial state with Gaussian noise. Their state spaces combine positions of body and joint with corresponding velocities. Control is achieved

by applying joint torques, which serve as the actions. Expert datasets are generated by first training policies through online reinforcement learning (Fu et al., 2020; Gallouédec et al., 2024) until high performance, then executing these policies to produce trajectories of states $\mathbf{x}_i$ and actions $\mathbf{y}_i$. Here, $\mathbf{x}_i$ encodes robot positions, joint angles, velocities, and angular velocities, while $\mathbf{y}_i$ denotes the applied joint torques.

An episode of expert demonstration has a length of 50 for Reacher, and it has a length of 1,000 for all other environments. Thus, by taking 20,000 data points from each expert dataset, the regression model learns from at least 20 complete trajectories to clone the expert's behavior. For evaluation, we retain a subset of the full validation dataset, keeping the number of data points at 20% of the training data size. For small datasets (1K) used in Figures 4, 9 and 10, the test datasets contain 1,000 unseen samples.

There is no absolute threshold for what constitutes "low-data" versus "high-data," as this depends on problem complexity and data structure. In our experiments, we operationally define this distinction using datasets of 1,000 samples versus 20,000 samples—a 20-fold difference that produces qualitatively different generalization behavior. In the low-data regime, the training set provides sparse coverage of the true data manifold, introducing sampling artifacts such as spurious correlations and outlier effects that do not reflect the true underlying distribution. Models trained on such small datasets, particularly without sufficient regularization, tend to memorize these sample-specific patterns rather than learning generalizable structure. In the high-data regime with 20,000 samples, the training set provides denser coverage, the effect of sampling artifacts diminishes, and the empirical distribution more closely approximates the true underlying distribution.

Table 2: All hyperparameter settings involved for experiments on MuJoCo datasets. Each figure employs a subset of possible hyperparameter combinations.

|  | Hyperparameter | Value |
|---|---|---|
| Model Architecture | Number of hidden layers | $\{1, 2, 3, 4, 5\}$ |
|  | Hidden layer dimension | $\{64, 128, 256, 512, 1024\}$ |
|  | Activation function | ReLU |
|  | Number of linear projection layer ($\mathbf{W}$) | 1 |
| Training | Epochs | $3 \times 10^5$ (20K-datasets) |
|  |  | $5 \times 10^6$ (1K-datasets) |
|  | Batch size | 4096 (20K-datasets) |
|  |  | 1000 (1K-datasets) |
|  | Optimizer | SGD |
|  | Learning rate | $1 \times 10-2$ |
|  | Weight decay | $\{0, 1e^{-5}, 1e^{-4}, 3e^{-4}, 5e^{-4}, 7e^{-4}, 1e^{-3}, 3e^{-3}\}$, Reacher |
|  |  | $\{0, 1e^{-5}, 1/3/5/7e^{-4}, 1/3/5/7e^{-3}, 1e^{-2}, 3e^{-2}\}$, Otherwise |
|  | Seed | 0 |
|  | Compute resources | NVIDIA A100 40GB |
|  | Number of CPU compute workers | 4 |
|  | Requested compute memory | 16 GB |
|  | Average training time per model | 20 hours |

Table 2 summarizes all model hyperparameters and experimental settings for MuJoCo datasets. A subset of possible hyperparameter combinations is used for each figure:

- Figure 1 plots the min-max normalized Test MSE as a function of the min-max normalized NRC1 values for the model architecture 3-256 (3 hidden layers and 256 hidden units) and all possible weight decay values.

- Figure 3 establishes the relationship between NRC1 and $ID_H$. Each subplot includes all weight decays listed in Table 2. And each weight decay is combined with 9 model architectures in {3-64, 3-128, 3-256, 3-512, 3-1024, 1-256, 2-256, 4-256, 5-256}.

- Figs. 4, 9 and 10 empirically reveal how generalization ability is affected by $ID_H$. We focus on a single model depth of 3, and vary the model width among $\{64, 128, 256, 512, 1024\}$. For each model architecture, we evaluate all possible weight decay values listed in Table 2.

- Figure 6 depict how intrinsic dimension evolves for each network layer. The model architecture is fixed at 3-256 and the title of each subplot annotates the weight decay value.

- Figure 7 and 8 follow the same experimental setup as Figs. 4, 9 and 10, but emphasize on the comparison between $ID_H$ and $ID_P$.
- Figure 11 record NRC1 values along the training process. The model architecture is fixed to 3-256 for all datasets. And we show 10 weight decay values in $\{0, 0.0001, 0.0003, 0.0005, 0.0007, 0.001, 0.003, 0.005, 0.007, 0.01\}$.

## A.2 MNIST/CIFAR10 EXPERIMENTS

The regression models for both the MNIST and CIFAR-10 tasks were trained across a spectrum of hyperparameters to thoroughly investigate the effects of architecture and regularization on the learned representations. The specific settings for model architecture, optimizer, and other training parameters are detailed in Table 4.

Table 3: All hyperparameter settings involved for experiments on MNIST and CIFAR-10 datasets.

|  | Hyperparameter | Value |
| --- | --- | --- |
| Model Architecture | Number of hidden layers | 3 |
|  | Hidden layer dimension | $\{32, 64, 128, 256, 512\}$ |
|  | Activation function | ReLU |
| Training | Epochs | 200 |
|  | Batch size | 64 |
|  | Optimizer | Adam |
|  | Learning rate | $\{1 \times 10^{-3}, 5 \times 10^{-3}\}$ |
|  | Weight decay (MNIST) | $\{0, 10^{-5}, 10^{-4}, 3 \times 10^{-4}, 5 \times 10^{-4}, 7 \times 10^{-4}, 10^{-3}, 3 \times 10^{-3}, 7 \times 10^{-3}\}$ |
|  | Weight decay (CIFAR-10) | $\{0, 10^{-5}, 10^{-4}, 3 \times 10^{-4}, 5 \times 10^{-4}, 7 \times 10^{-4}, 10^{-3}\}$ |
|  | Seed | 0 |
|  | Compute resources | NVIDIA A100 80GB |
|  | Average training time per model | 2 hours |

We define noisy-target tasks as tasks where the regression targets contain information that causes models to learn patterns that do not generalize to unseen data. The term "noise" here does not refer to random measurement error or label corruption, but rather to information that varies between semantically similar examples due to instance-specific characteristics rather than reflecting the true underlying function.

The synthetic MNIST regression task represents a low-noise setting. The feature extractor is a CNN trained specifically on MNIST until achieving over 99% classification accuracy, creating a self-consistent target-generation pipeline. During training, the CNN learns to discard instance-specific variations (such as stroke thickness, slight rotations, or pixel-level noise) that are irrelevant for predicting the target label, retaining only the semantically meaningful information. As a result, the mapping from MNIST images to projected features is smooth and well-aligned with the data domain, enabling models to generalize effectively (Figs. 4 (d),(g)).

In contrast, the synthetic CIFAR-10 regression task is a noisy-target setting due to domain mismatch. CIFAR-10 images are processed through a ResNet-18 pretrained on ImageNet to extract features, which are then projected to 10-dimensional targets. The ImageNet encoder captures fine-grained visual details—texture patterns, color distributions, edge structures—optimized for ImageNet's 1,000 classes. When applied to CIFAR-10's 32×32 images, these features encode not only semantic content but also instance-specific characteristics: two images of cars may differ in color, lighting, rendering artifacts, or background elements, all of which significantly influence the extracted features. Models can fit these instance-specific components during training, achieving low training error. However, at test time, new images from the same semantic classes exhibit different instance-specific details. The learned mappings for these non-generalizable components do not transfer, resulting in poor generalization. This is evident in Figs. 4 (b),(e), where models achieve low training MSE but high test MSE, characteristic of overfitting to target noise.

## B    RELATED WORK

**NC under varied settings on classification.** The phenomenon of neural collapse was first empirically observed by Papyan et al. (2020), who demonstrated its emergence during TPT in deep neural network models for classification tasks. Building on this empirical finding, researchers have developed theoretical frameworks to analyze NC, such as the Unconstrained Feature Model (UFM) (Mixon et al., 2020) and the layer-peeled model (Fang et al., 2021). Using these models, numerous studies have demonstrated that NC provably occurs under diverse conditions (Han et al., 2021; Tirer & Bruna, 2022; Yaras et al., 2022; Zhou et al., 2022a;b; Zhu et al., 2021) and using various loss functions such as label smoothing (Guo et al., 2024). See also (Hong & Ling, 2023; Thrampoulidis et al., 2022; Yang et al., 2022).

**NC beyond single-label classification.** Recent research has extended the principles of NC beyond its original single-label classification setting. Li et al. (2023a) demonstrated that in multi-label classification, embeddings reside within the linear span of their label means. Andriopoulos et al. (2024) generalized NC to neural multivariate regression, formalizing it as Neural Regression Collapse (NRC). Ma et al. (2025) showed that NC emerges in deep ordinal regression, analyzing it through the UFM framework. In large language models, Wu & Papyan (2024) identified a "linguistic collapse". Súkeník et al. (2025) proved that NC represents the globally optimal configuration in modern deep regularized architectures, including ResNets and transformers.

**Intrinsic dimension in deep neural networks.** Several works investigate the intrinsic dimension of data manifolds and representations in deep neural networks (Denil et al., 2013; LeCun et al., 1989). Classical methods estimate intrinsic dimension from local neighborhoods (Allegra et al., 2020; Amsaleg et al., 2015; Facco et al., 2017; Levina & Bickel, 2004), which have been extended to neural settings. Ma et al. (2018a) show that local intrinsic dimension (LID) can distinguish adversarial from natural image data. More recently, Yin et al. (2024) focused on per-sample LID to identify when LLMs produce untruthful outputs.

A parallel line of research uses tools from topological data analysis to study neural networks. Some works analyze the trained network by constructing topological invariants from layer weights, such as Neural Persistence (Rieck et al., 2018), which can distinguish between models that overfit or generalize well. Others analyze the underlying graph structure of networks (Corneanu et al., 2019; 2020). While often empirical, these approaches provide a novel perspective on network properties. More recent work (Birdal et al., 2021) has begun to place these topological methods on a firmer theoretical foundation using statistical persistent homology.

Beyond empirical estimations, intrinsic dimension has been studied as a measure of model complexity. Recent approaches analyze the degrees of freedom in parameter space (Gao & Jojic, 2016; Janson et al., 2015), compressibility via pruning (Blier & Ollivier, 2018), and intrinsic dimension (Ansuini et al., 2019; Li et al., 2018; Ma et al., 2018b; Pope et al., 2021). Compression-based generalization bounds (Arora et al., 2018; Barsbey et al., 2021; Hsu et al., 2021; Suzuki et al., 2018; 2019) have shown that networks that can be represented in a lower-dimensional space exhibit lower generalization error. See also Simsekli et al. (2020); Birdal et al. (2021); Zhu et al. (2018).

**Comparison between Regression and Classification** Previous work on manifold learning for neural classification has demonstrated that the intrinsic dimension of the last hidden layer is negatively correlated with generalization ability. In particular, models achieving lower intrinsic dimension in the penultimate layer were found to exhibit superior test accuracy, with the lowest intrinsic dimension-model attaining the highest top-5 accuracy, see Section 3.2 and Figure 4 in Ansuini et al. (2019). Additionally, Papyan et al. (2020) connect neural collapse to robust decision boundaries, Galanti et al. (2021) demonstrate that collapse patterns improve few-shot and transfer learning, and Li et al. (2022) show the degree of collapse in downstream representations strongly predicts transfer accuracy. Complementing these empirical results, there are also theoretical results showing the benefits of neural collapse for classification (Gao et al., 2023; Wang & Palmer, 2023; Hui et al., 2022).

In regression, however, our findings indicate a more nuanced picture. We demonstrated the existence of a "soft" threshold at $ID_Y$, which delineates distinct generalization regimes. In the under-compressed regime with low-data tasks and high-noise tasks, reducing $ID_H$ improves generalization, consistent with the monotonic complexity-performance paradigm observed in classification. However, in the over-compressed regime and in the under-compressed regime with high-data tasks and

low-noise tasks, the opposite holds: increasing $ID_H$ improves generalization, a phenomenon absent in classification tasks. Thus, in regression, generalization performance depends non-monotonically on the relationship between the learned feature manifold and the intrinsic dimension of the targets.

## C    INTRINSIC DIMENSION AND THE 2-NN ALGORITHM

The intrinsic dimension (ID) of a dataset is the minimum number of coordinates needed to represent the data faithfully. If data points lie on or near a $d$-dimensional manifold $\mathcal{M}$ embedded in $\mathbb{R}^D$ with $d \ll D$, then $d$ is the intrinsic dimension. For example, a circle in 3D space has $d = 1$ and a sphere surface in 10D has $d = 2$.

A critical distinction exists between PCA dimensionality and intrinsic dimension. Consider a 1D spiral embedded in $\mathbb{R}^{10}$ parameterized by $t$. The spiral winds through space with substantial variance across all 10 axes, requiring multiple PCA components to capture the signal. Yet, the intrinsic dimension is exactly $d = 1$, because specifying a single scalar value, such as the arc length from the origin, is sufficient to uniquely locate any point on the curve. This illustrates that curved or folded manifolds can require many linear directions to approximate while having low intrinsic dimension. Figure 2 demonstrates this for neural regression: collapsed features lie near a 2D linear subspace (yellow plane) yet occupy a nonlinear 1D manifold within it.

We estimate ID using the 2-NN estimator (Facco et al., 2017), which exploits a fundamental geometric property: in a $d$-dimensional space, the probability of finding neighbors within a given distance scales with dimension $d$. The key insight is to consider not absolute distances, but the ratio $\mu = r_2/r_1 \geq 1$ of the second to first nearest-neighbor distances. Remarkably, under the assumption of locally uniform density (density approximately constant within the range of the second neighbor), this ratio has a distribution that depends only on the intrinsic dimension $d$, with the local density completely canceling out. Specifically, the cumulative distribution function of $\mu$ is

$$F(\mu) = 1 - \mu^{-d}, \quad \mu \geq 1$$

This property makes the estimator robust to density variations, since we never need to estimate the density itself. Taking logarithms yields the linear relationship

$$\log(1 - F(\mu)) = -d \log \mu$$

The 2-NN algorithm estimates $d$ by computing $\mu_i$ for each point, constructing the empirical CDF $F_{\text{emp}}$, and performing linear regression on the transformed coordinates $\{(\log \mu_i, -\log(1 - F_{\text{emp}}(\mu_i)))\}$. The requirement of local uniformity only within the second-neighbor distance is much weaker than global uniformity, making the estimator practical for real datasets with varying density and curvature.

---

**Algorithm 1:** 2-NN Intrinsic Dimension Estimation

---

**Input:** Dataset $\mathcal{X} = \{\mathbf{x}_i\}_{i=1}^M$.
**Output:** Estimated intrinsic dimension $\hat{d}$.
**for** $i \leftarrow 1$ **to** $M$ **do**
  Compute Euclidean distances to the first and second nearest neighbors, $r_1(\mathbf{x}_i)$ and $r_2(\mathbf{x}_i)$;
  Compute ratio $\mu_i \leftarrow r_2(\mathbf{x}_i)/r_1(\mathbf{x}_i)$;
Sort the ratios such that $\mu_{\sigma(1)} \leq \mu_{\sigma(2)} \leq \cdots \leq \mu_{\sigma(M)}$;
**for** $i \leftarrow 1$ **to** $M$ **do**
  Assign empirical CDF value $F_{emp}(\mu_{\sigma(i)}) \leftarrow \frac{i}{M}$;
Construct the coordinate set for regression:

$$\mathcal{S} \leftarrow \left\{ \left( \log \mu_{\sigma(i)}, -\log\left(1 - F_{emp}(\mu_{\sigma(i)})\right)\right) \right\}_{i=1}^{M-1}$$

Fit a line through the origin to $\mathcal{S}$ using least squares;
**return** Slope of the fitted line ($\hat{d}$);

---

# D    EVOLUTION OF INTRINSIC DIMENSION DURING TRAINING

To further understand the behavior of collapsed models and their counterparts, we track the evolution of the intrinsic dimension throughout training and provide insights. Figure 6 provides illustrative examples for a collapsed and a non-collapsed model:

- For both the collapsed and non-collapsed models, the intrinsic dimension of the last-layer features invariably decreases monotonically until convergence.

- For the collapsed model, the deeper the layer in the network, the lower the intrinsic dimension at the end of training. ReLU activations cause a mild reduction in intrinsic dimension in comparison with the reduction in intrinsic dimension between consecutive layers (ignoring ReLU). Notably, the final intrinsic dimension of the output layer, which gives the actual vector-valued predictions, can be significantly lower than $ID_H$.

- For non-collapsed models, we usually see — but not always — $ID_H$ decrease monotonically as we move from shallow to deep layers. Furthermore, we observe that during training, the intrinsic dimension of the output layer hugs the intrinsic dimension of the targets. Thus, tracking the intrinsic dimension of the output layer provides yet another criterion for discriminating between collapsed and non-collapsed models; see Appendix E.

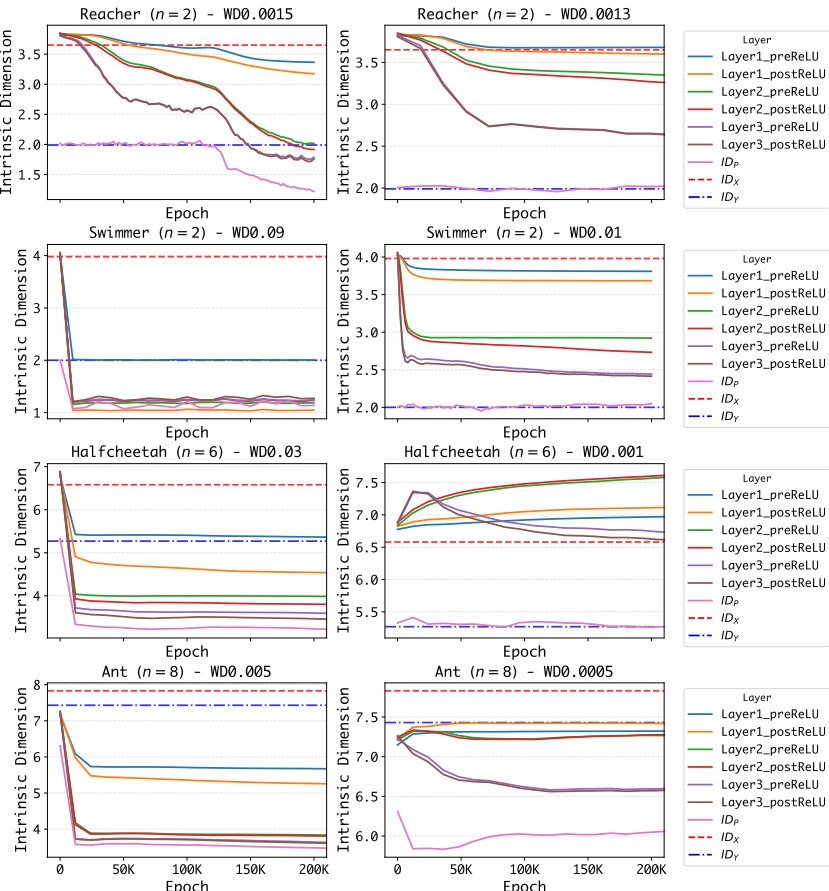

Figure 6: Intrinsic dimension of input, output, and hidden layers over training epochs for a collapsed (left) and a non-collapsed model (right) for MuJoCo datasets. Each subfigure shows the evolution of intrinsic dimension across layers with blue, red dashed, and pink lines denoting the intrinsic dimension of inputs, targets, and predicted outputs, respectively.

# E   INTRINSIC DIMENSION AND OUTPUT LAYER

We consider here the intrinsic dimension of the outputs (equivalently, the final predictions), $ID_P$. We will see that here too the relationship between intrinsic dimension and generalization exhibits key differences between classification and regression.

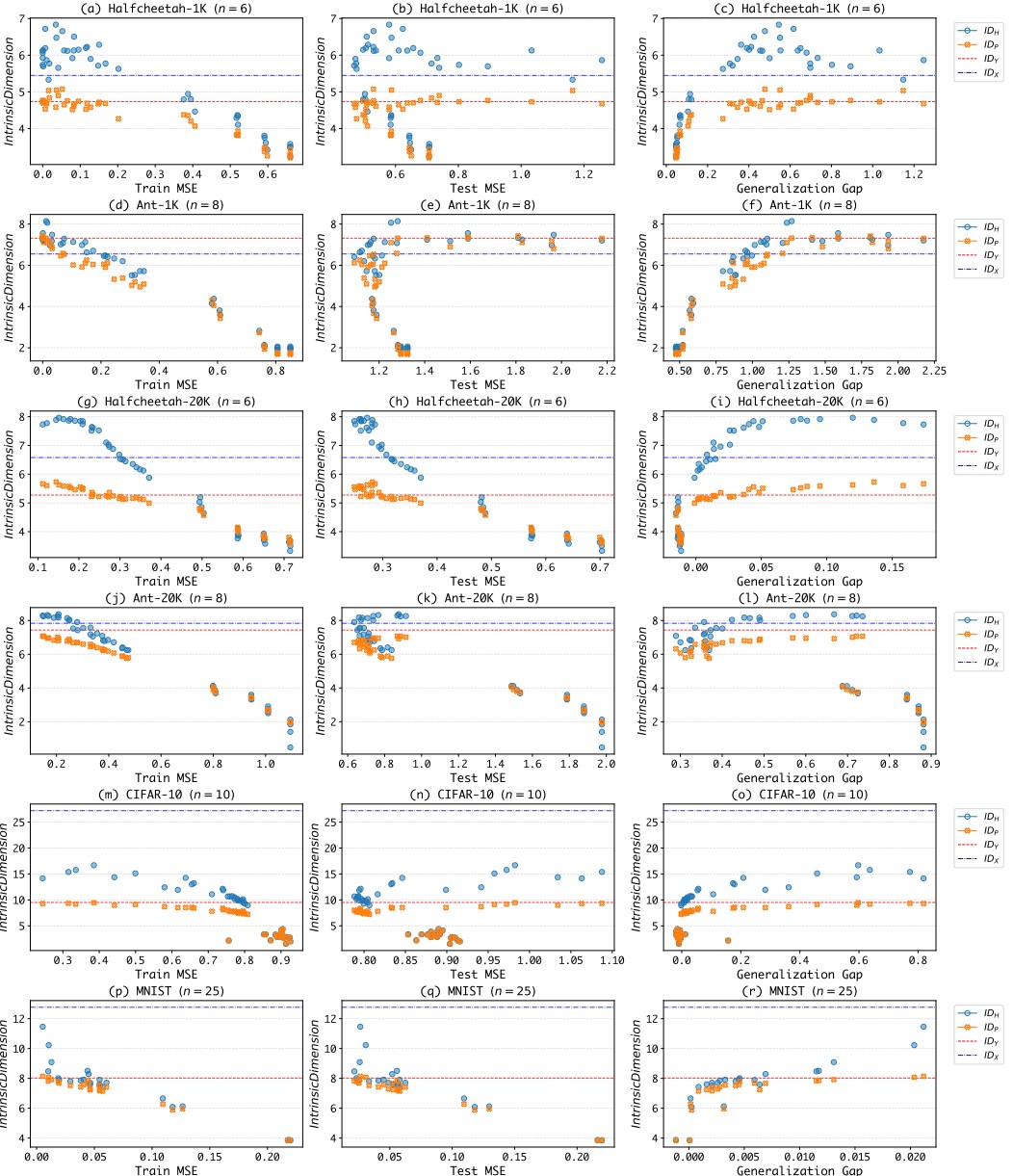

Figure 7: Comparison between $ID_H$ and $ID_P$ for Halfcheetah, Hopper, CIFAR-10, and MNIST datasets

With respect to the output layer, a structural constraint arises from the classification setting. Specifically, the intrinsic dimension of the output layer necessarily satisfies

$$\log_2 C \le ID_P \le C,$$

where $C$ is the number of classes. Empirical results consistently show $ID_P$ equals the lower bound of this inequality if the model generalizes well. We refer the reader to the discussion in Section 3.1 of Ansuini et al. (2019). Conversely, saturation of the upper bound, i.e., $ID_P \simeq C$, is associated with

poor generalization performance, suggesting that maximal output layer dimensionality corresponds to overfitting in classification tasks, see Section 3.5 in Ansuini et al. (2019).

In contrast, for neural multivariate regression, the structure of the output leads to the trivial bound

$$1 \leq ID_P \leq n,$$

where $n$ is the number of output variates. Interestingly, our empirical findings reveal a departure from the classification setting. As shown in the middle column in Figures 7-8, when the test MSE is low, the intrinsic dimension of the output layer, $ID_P$ satisfies $ID_P \simeq ID_Y$, which can be close to $n$, saturating the upper bound of the inequality above. Notably, unlike in classification, this saturation is associated with improved test performance. By contrast, when $ID_P$ falls below $ID_Y$, test MSE performance deteriorates, see Figures 7-8.

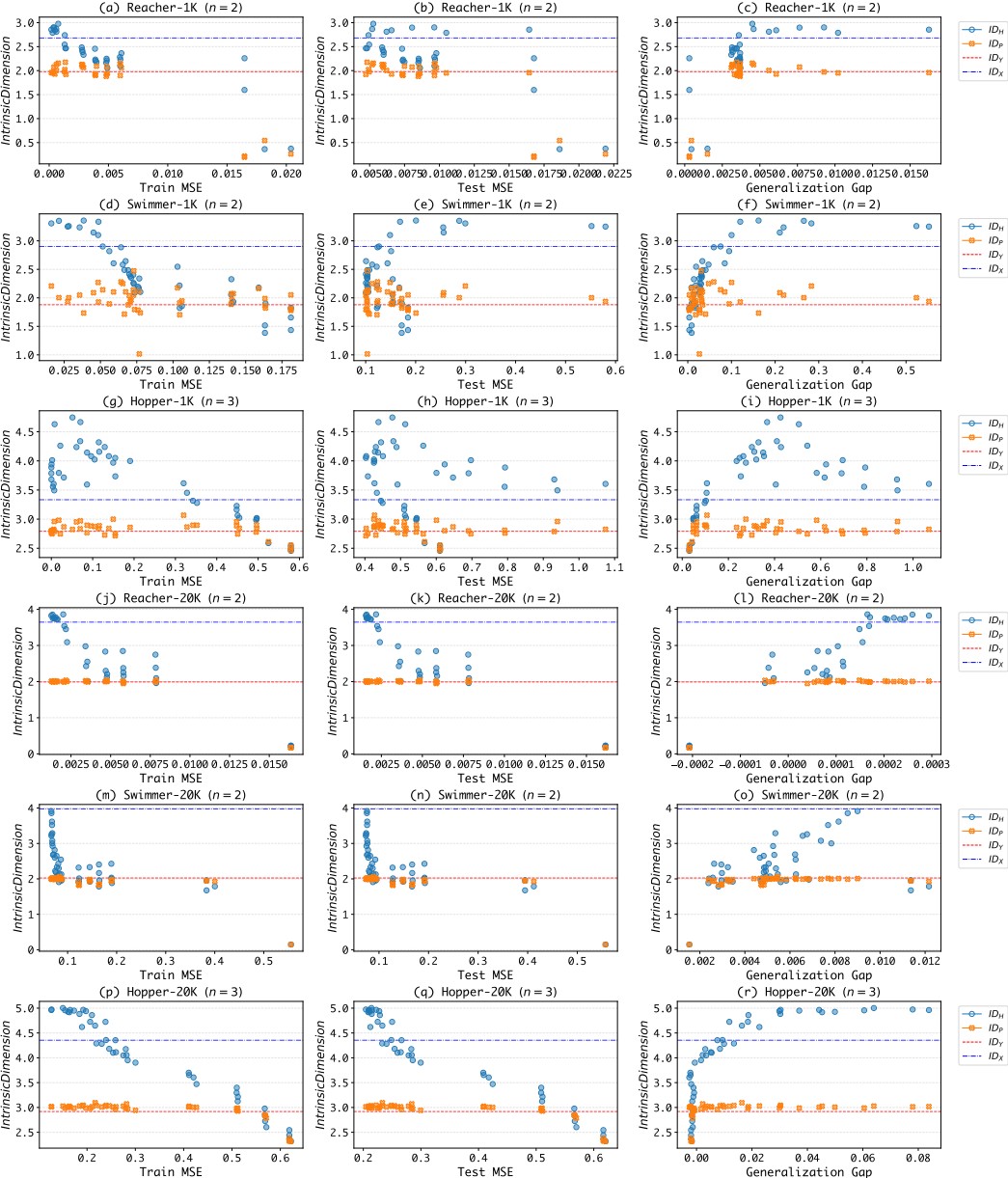

Figure 8: Comparison between $ID_H$ and $ID_P$ for Reacher, Swimmer and Ant datasets

# F    ADDITIONAL EXPERIMENTS ON GENERALIZATION

This section lists additional results that complement the experiments in the main body for all considered datasets. Figure 9 and 10 shows how generalization power correlates with $ID_H$. Our key takeaways are summarized in Table 4.

Table 4: Key Takeaways for Generalization.

| Regime | ID | Typical behavior |
|---|---|---|
| Over-compressed | $ID_H < ID_Y$ | Underfitting with large train and test MSE |
| Balanced | $ID_H \approx ID_Y$ | Sweet spot in low-data and noisy tasks |
| Under-compressed | $ID_H \gg ID_Y$ | Benign overfitting with enough low-noise data |

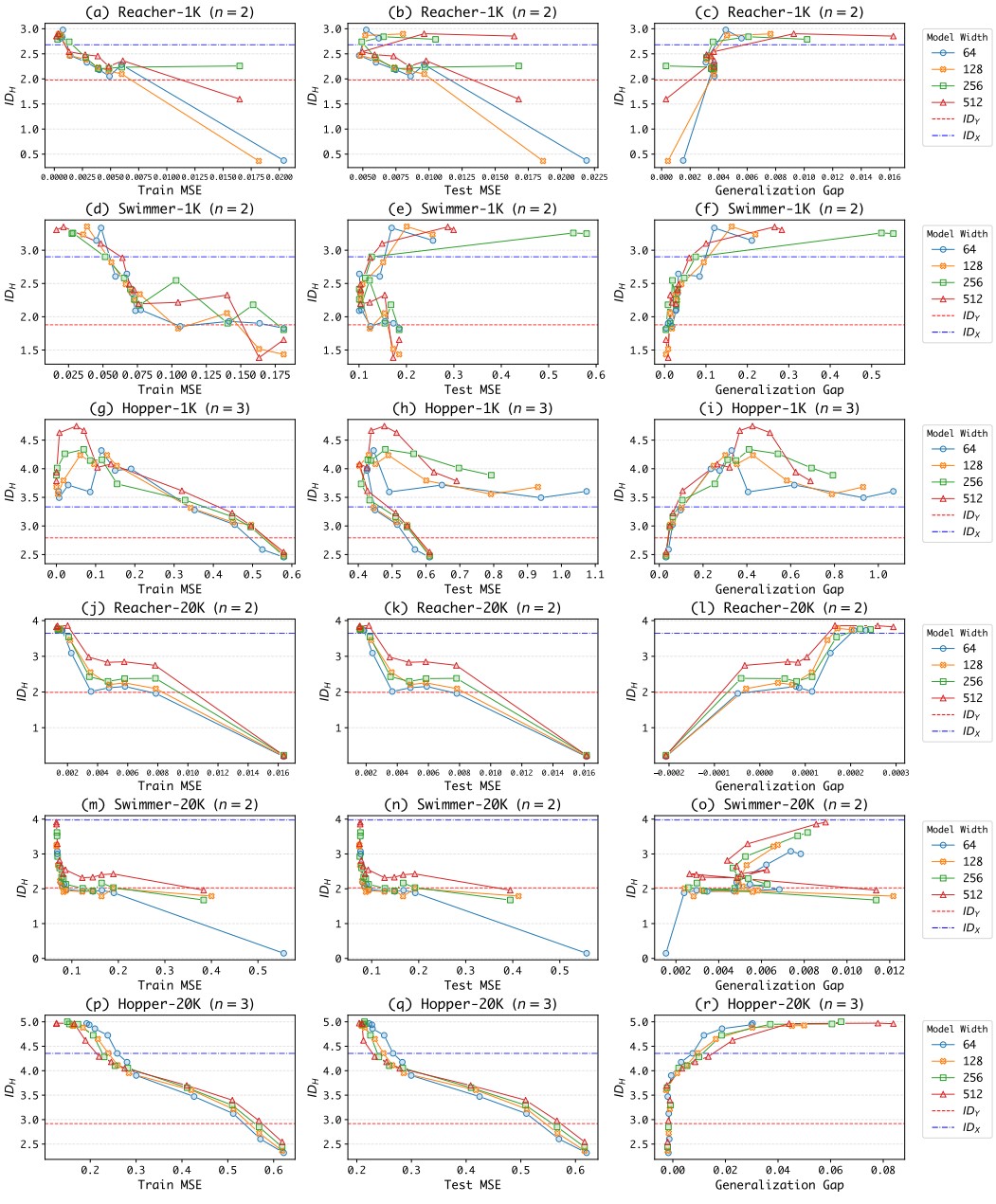

Figure 9: Generalization ability and Intrinsic Dimension for Reacher, Swimmer, and Hopper datasets.

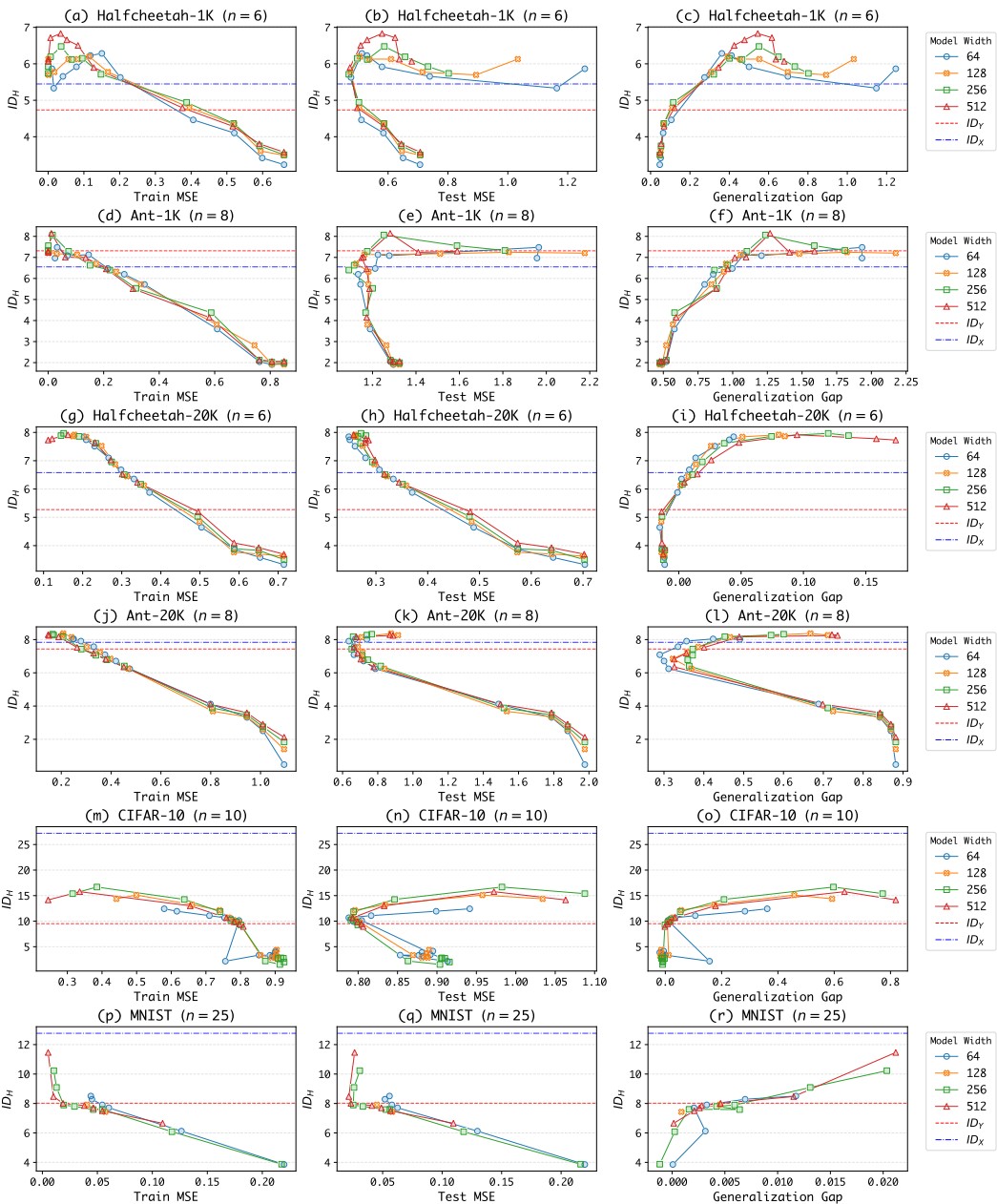

Figure 10: Generalization ability and Intrinsic Dimension for Halfcheetah, Ant, CIFAR-10, and MNIST datasets.

## G    HOW DOES REGULARIZATION AFFECT NRC AND INTRINSIC DIMENSION?

### G.1    WEIGHT DECAY

Weight decay is a canonical and widely adopted model regularization technique for preventing large models from overfitting to data. Real-world applications include but are not limited to (1) regularizing transformer backbones for large language models (Wolf et al., 2020) and robotic generalist policies (Chen et al., 2021); (2) participating, by default, in the common Pytorch implementation of AdamW optimizer (Loshchilov & Hutter, 2019; PyTorch Contributors, 2025) with `weight_decay=0.01`; (3) improving sample efficiency of online reinforcement learning algorithms (Liu et al., 2021; Li et al., 2023b); (4) and facilitating research in model plasticity in deep learning (Lyle et al., 2023; Nauman et al., 2024a; Ceron et al., 2024a).

Figure 11 investigates NRC1 for values of the weight decay parameter $\lambda_{WD}$. We see that when $\lambda_{WD}$ is zero or small, there is no neural regression collapse; but if we increase the weight decay, the NRC1 geometric structure quickly emerges during training. This matches the empirical observation in Andriopoulos et al. (2024).

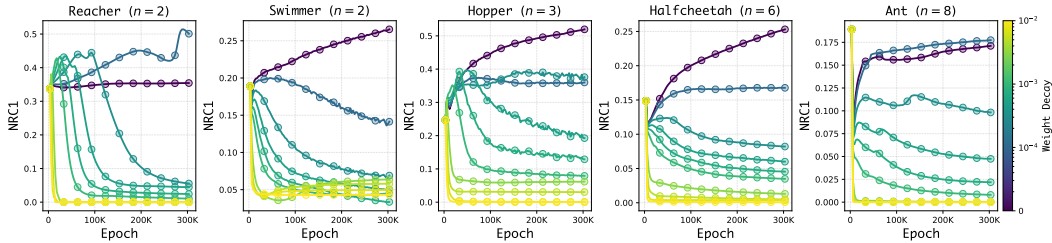

Figure 11: NRC1 decreases as weight decay becomes stronger, leading to model collapse.

### G.2    DROPOUT REGULARIZATION

Modern implementations, across deep learning domains, continue to rely on dropout regularization (Srivastava et al., 2014) to mitigate overfitting, underscoring its persistent role in practical training, such as computer vision (Dosovitskiy et al., 2021), NLP (Devlin et al., 2019; Wolf et al., 2020), and reinforcement learning (Hiraoka et al., 2022). We empirically analyze how dropout regularization influences neural regression collapse by varying its strength from a large range ($\in \{0, 0.0001, 0.0005, 0.001, 0.005, 0.01, 0.05, 0.1, 0.2, 0.3, 0.4, 0.5, 0.6, 0.7, 0.8\}$). *No* weight decay is applied. In this section, datasets include Hopper, Halfcheetah, and Ant with two sizes, 1K (Figure 12) and 20K (Figure 13). The horizontal red dashed line represents $ID_Y$.

Figure 12(a) and Figure 13(a) show the relationship between $ID_H$ and NRC1. We first confirm the same conclusion as made in Section 5, despite the new regularization. $ID_H$ provides a more refined geometric structure than NRC1. Collapsed models with near-zero NRC1 values have varying $ID_H$ below or in the vicinity of $ID_Y < n$, while non-collapsed models with non-trivial NRC1 maintain their $ID_H$ to be above $ID_Y$ and to be positively correlated with $NRC1$. Interestingly, dropout regularization differs from weight decay in that mild dropout (e.g., $\leq 0.01$) can effectively prevent models from collapse by increasing both NRC1 and $ID_H$. This observation sheds light on a geometric interpretation of the effectiveness of mild dropout in reinforcement learning as proposed by Hiraoka et al. (2022).

Figure 12(b) and Figure 13(b) show the relationship between $ID_H$ and test MSE. The results again verify the three regimes discussed in Section 6. For both data sizes, models over-compress features when $ID_H < ID_Y$ and thus lead to increasing test MSE (and thus poor generalization). Then, for small datasets with 1K samples, $ID_H \approx ID_Y$ identifies the sweet spot where test MSE tends to be the lowest and exhibits the 'U-shape' plots. Note that models trained with Hopper datasets have not collapsed yet, so they only exhibit the upper part of the 'U-shape'. Finally, with more samples, e.g., 20K, $ID_H \gg ID_Y$ achieves the best generalization with under-compressed models.

In summary, dropout regularization offers an alternative approach to adjusting the degree of model collapse, while all conclusions drawn from the main body remain intact and inclusive. In addition,

mild dropout regularization is more effective than weight decay regularization in increasing NRC1 and $ID_H$ metrics for the under-compressed regime.

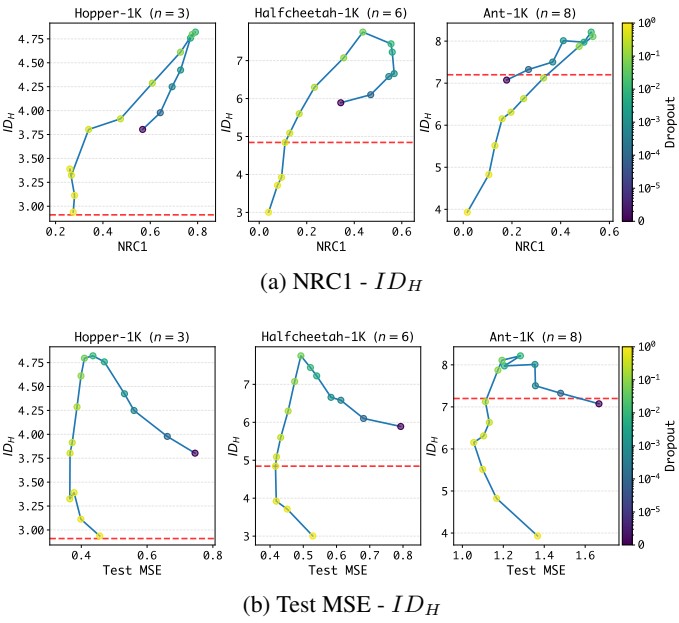

Figure 12: Relationship between $ID_H$ and NRC1 and Test MSE for Hopper-1K, Halfcheetah-1K, and Ant-1K datasets, when applying model dropout regularization. The horizontal red dashed line represents $ID_Y$.

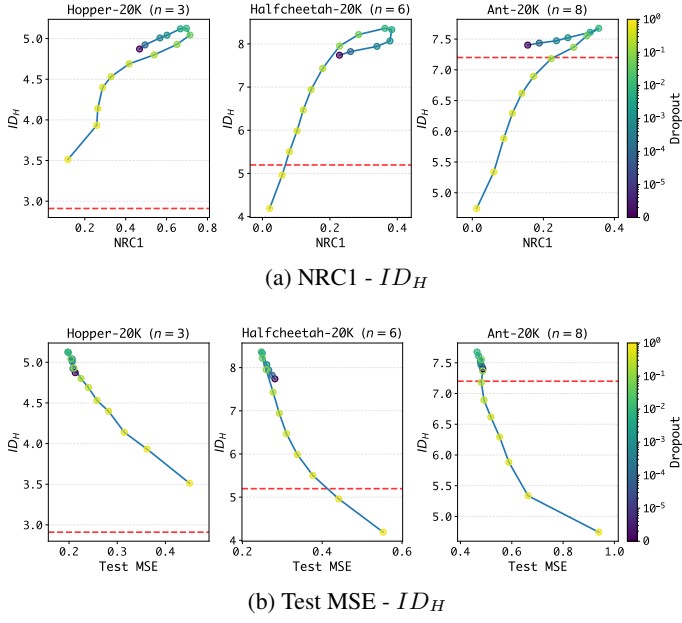

Figure 13: Varying model dropout regularization, relationship between $ID_H$ and NRC1 and Test MSE for Hopper-20K, Halfcheetah-20K, and Ant-20K datasets. The horizontal red dashed line represents $ID_Y$.

## G.3 MODEL DEPTH

With mild weight decay regularization, we find that increasing model depth leads to smaller NRC1 and $ID_H$ and thus to more collapsed features. In Figure 14, we examine the relationship between $ID_H$ and NRC1 for Hopper-20K and Halfcheetah-20K datasets. We fix the model width to be 256 and vary the model depth ($\in \{2, 3, 4, 5\}$). For each model depth, we show five mild weight decay values: 0.0001, 0.0003, 0.0005, 0.0007, 0.001. The figure shows that increasing model depth gradually pushes points to the region on the bottom left, where both NRC1 and $ID_H$ are small. For example, Halfcheetah-20K with a depth of 5 can result in collapsed models with $ID_H < ID_Y$.

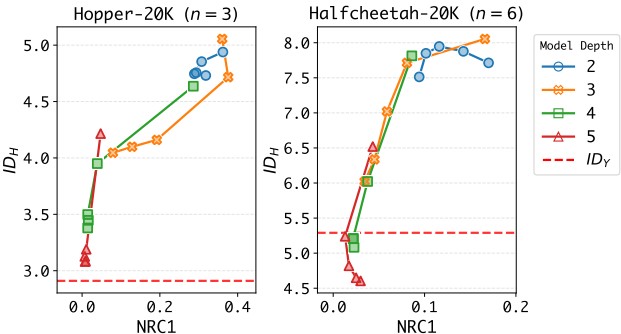

Figure 14: Relationship between $ID_H$ and NRC1 for Hopper-20K and Halfcheetah-20K datasets, when varying model depth. All models have a hidden size of 256. The horizontal red dashed line represents $ID_Y$.

# H EMPIRICAL ANALYSIS ON MORE CHALLENGING TASKS

We extend our empirical analysis to more challenging tasks with varying data sizes, increased intrinsic dimensions, and visual inputs.

Table 5: Overview of additional datasets employed in this section.

| Dataset | Data Size | Input Type | Input Dim ($D$) | Input ID ($ID_X$) | Target Dim ($n$) | Target ID ($ID_Y$) |
|---|---|---|---|---|---|---|
| Humanoid | 500,000 | raw state | 348 | 11.02 | 17 | 9.85 |
| Relocate | 500,000 | raw state | 39 | 6.90 | 30 | 19.82 |
| Cheetah_run | 80,000 | RGB image | $84 \times 84 \times 9^3$ | 8.26 | 6 | 6.00 |
| Humanoid_walk | 100,000 | RGB image | $84 \times 84 \times 9$ | 9.41 | 21 | 14.84 |

**Humanoid (MuJoCo locomotion)**    The Humanoid dataset (Younis et al., 2024) is generated from the MuJoCo physics simulator, introduced in the main body and Appendix A.1. Each state consists of high-dimensional proprioceptive information, and the corresponding targets are the expert control torques applied at each joint. The goal is to enable the humanoid to run forward stably while maintaining balance, which is substantially more difficult than all previously considered MuJoCo tasks due to its high degrees of freedom and complex contact dynamics. Among all MuJoCo environments, Humanoid is widely regarded as the most challenging.

**Relocate (Adroit manipulation)**    The Relocate dataset (Fu et al., 2020) comes from the Adroit suite (Rajeswaran et al., 2018) of dexterous manipulation tasks. Adroit uses a simulated 24 degrees of freedom (24-DoF) robotic hand, combined with an arm of up to 6-DoF, with rich contact and articulation dynamics. In the Relocate task, the state includes joint positions, velocities, hand pose information, and kinematic information about the ball and target. The action corresponds to the joint torques for the 24 actuators and to the arm movement. The objective is to grasp a small object and

---

[3]Frame stack is commonly applied for visual control tasks. A single observation is of shape $84 \times 84 \times 3$. A frame stack of 3 is used in our experiments, resulting in visual inputs of dimension $84 \times 84 \times 9$.

relocate it to a specified target position. It is a long-horizon manipulation task requiring precise coordination and contact control. Among the four Adroit tasks in the D4RL benchmark (Fu et al., 2020), Relocate is widely considered the most difficult due to its combination of dexterity, precision, and exploration complexity.

**Cheetah_run & Humanoid_walk (Visual continuous control)**  The Cheetah_run and Humanoid_walk datasets (Lu et al., 2023) are visual control benchmarks constructed from demonstrations generated in the DeepMind Control Suite (Tassa et al., 2018). Inputs consist of raw image observations (e.g., 84×84x3 RGB frames), and targets correspond to the continuous control commands. In this section, the Cheetah_run dataset consists of expert demonstrations, as is the case for all previous datasets, while the Humanoid_run dataset contains some noisy suboptimal behavior in addition to the expert demonstrations ('medium-expert dataset'). This adds difficulty in extracting a good policy by imitating the dataset's behavior. We use a CNN image encoder (consisting of 4 `conv2d` layers and ReLU activation), followed by a 3-layer MLP policy network.

Visual control is particularly interesting and challenging because each frame provides only partial information about the system state, effectively forming a Partially Observed Markov Decision Process (POMDP) (Yarats et al., 2022). As a result, the model must infer underlying transition dynamics and identify salient visual features directly from high-dimensional pixel inputs. Within the visual D4RL tasks (Lu et al., 2023), Humanoid_walk is the most challenging due to the humanoid's instability and high-dimensional dynamics, whereas Cheetah_run is comparatively easier but still nontrivial.

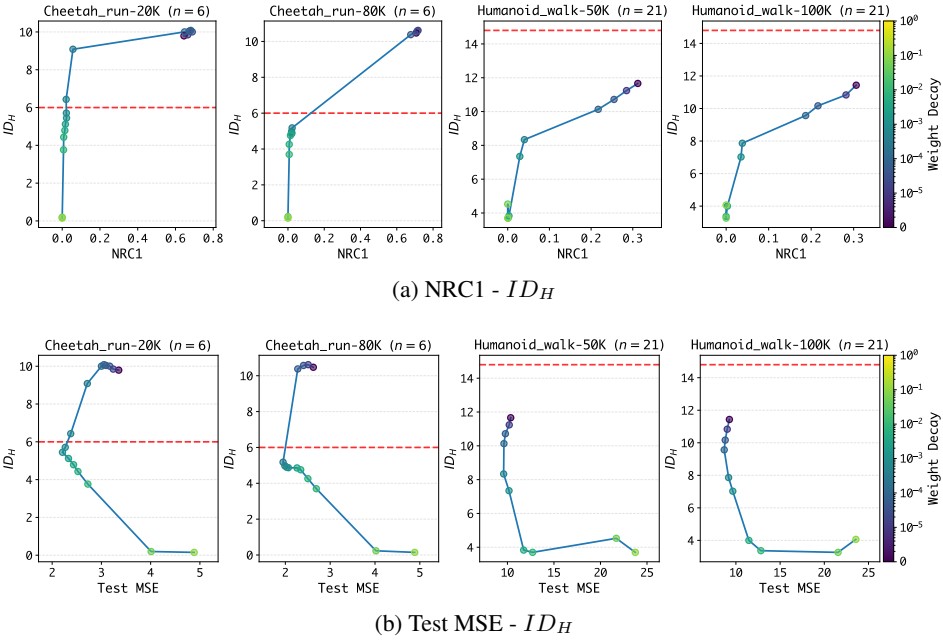

(a) NRC1 - $ID_H$

(b) Test MSE - $ID_H$

Figure 15: Relationship between $ID_H$ and NRC1 and Test MSE for Cheetah_run-20K, Cheetah_run-80K, Humanoid_walk-50K and Humanoid_walk-100K. The horizontal red dashed line represents $ID_Y$.

### H.1   VISUAL CONTROL TASKS

Figure 15(a) shows the relationship between $ID_H$ and NRC1 for the visual control datasets with varying sizes. Consistent with the conclusions in Section 5, $ID_H$ provides a more refined geometric structure than NRC1. Collapsed models with small NRC1 values have varying $ID_H$ below or in the vicinity of $ID_Y < n$, while non-collapsed models with non-trivial NRC1 maintain their $ID_H$ to be above $ID_Y$ and to be positively correlated with NRC1. Notably, for the most challenging Humanoid_walk task, which has a substantially higher $ID_Y$ due to its complex high-dimensional dynamics, models trained with *zero* weight decay initially exhibit neural regression collapse with $ID_H < ID_Y$. This explains the extremely large test MSE observed for Humanoid_walk in Figure 15(b) and its negative

correlation with $ID_H$, which matches the over-compressed regime ($ID_H < ID_Y$) summarized in Table 4. In comparison, the two Cheetah_run datasets exhibit the 'U-shape' in the relationship between $ID_H$ and test MSE. This emphasizes the sweet spot with $ID_H \approx ID_Y$, which approaches the lowest test MSE in noisy and challenging tasks.

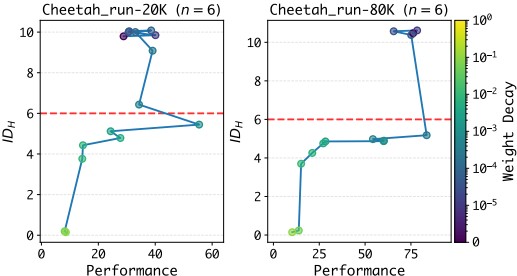

Figure 16: Relationship between $ID_H$ and normalized model performance ($\in [0, 100]$; the higher, the better) by training via behavior cloning on Cheetah_run-20K, and Cheetah_run-80K datasets. We evaluate the trained models to execute locomotion tasks in the DeepMind Control Suite (DMC) robotic simulation (Tassa et al., 2018). The horizontal red dashed line represents $ID_Y$.

**Practical guideline in control tasks**    Our empirical results have direct implications for evaluating real-world control models, where evaluation is often expensive, unsafe, and risky (Levine et al., 2020). Validation MSE provides little indication of actual control performance in the real tasks. Practitioners must therefore rely on costly real-environment interactions to tune hyperparameters and assess the policy. In Figure 16, we show that the relationship between $ID_H$ and $ID_Y$ not only predicts test MSE but also aligns with the *true* control performance obtained from environment interaction. In particular, the relationship between $ID_H$ and normalized model performance exhibits the opposite 'U-shape' behavior, achieving its best score when $ID_H \approx ID_Y$, mirroring the sweet-spot identified in Section 6.

In this way, intrinsic dimension provides a lightweight surrogate for policy evaluation, reducing reliance on frequent, costly, and potentially unsafe real-environment testing. During training, one can monitor the relationship between $ID_H$ and $ID_Y$ to narrow down the hyperparameter search by avoiding the $ID_H < ID_Y$ scenario. When $ID_H \geq ID_Y$, the correlation between $ID_H$ and the test MSE can also reflect its correlation with the real performance.

### H.2    RELOCATE & HUMANOID

**Humanoid**    In Figure 17, we show 'NRC1 - $ID_H$' and 'Test MSE - $ID_H$' plots for Humanoid datasets with varying sizes. In Figure 17(a), we observe that $ID_H$ provides a more refined geometric structure than NRC1. Collapsed models with near-zero NRC1 values have varying $ID_H$ below or in the vicinity of $ID_Y < n$, while non-collapsed models with non-trivial NRC1 maintain their $ID_H$ to be above $ID_Y$ and to be positively correlated with NRC1. Then, Figure 17(b) again verifies the three regimes discussed in Section 6. For all data sizes, models over-compress features when $ID_H < ID_Y$ and thus lead to increasing test MSE (and thus poor generalization). Then, for small datasets with 1-50K samples, $ID_H \approx ID_Y$ identifies the sweet spot where test MSE tends to be the lowest and exhibits the 'U-shape' plots. Finally, with more samples, e.g., 100-500K, $ID_H > ID_Y$ achieves the best generalization with under-compressed models.

**Relocate**    For the Relocate task with varying data sizes, which have the highest $ID_Y$ due to its complex task dynamics, models trained with *zero* weight decay initially exhibit a severe collapse with both NRC1 $\approx 0$ and $ID_H \ll ID_Y$. Unsurprisingly, Test MSE remains large for all data sizes and decay values, and it also negatively correlates with $ID_H$, uncovering the over-compressed regime.

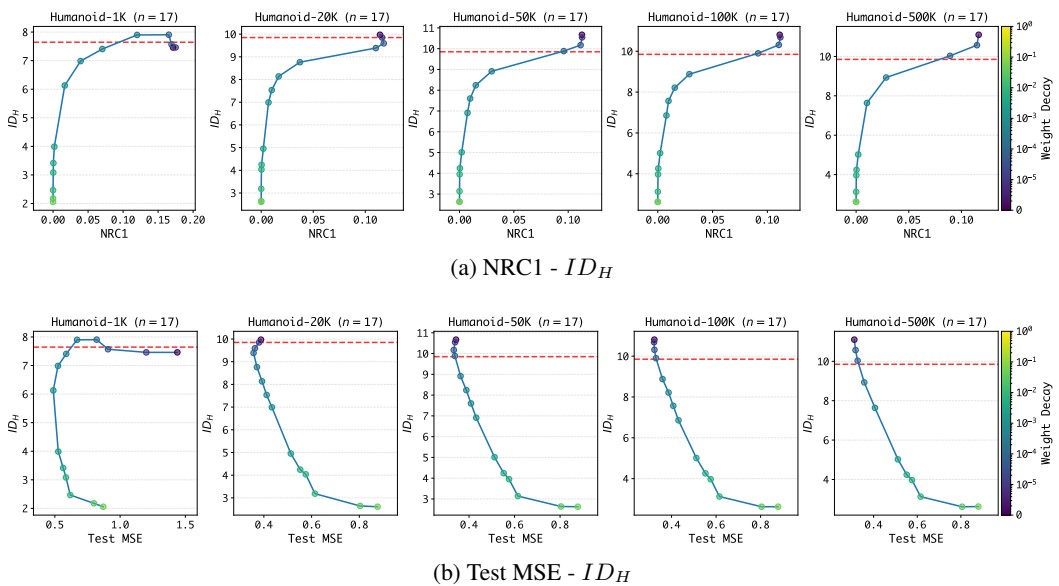

Figure 17: Relationship between $ID_H$ and NRC1 and Test MSE, for Humanoid locomotion task, when varying data size ($\in \{1K, 20K, 50K, 100K, 500K\}$). The red dashed line represents $ID_Y$.

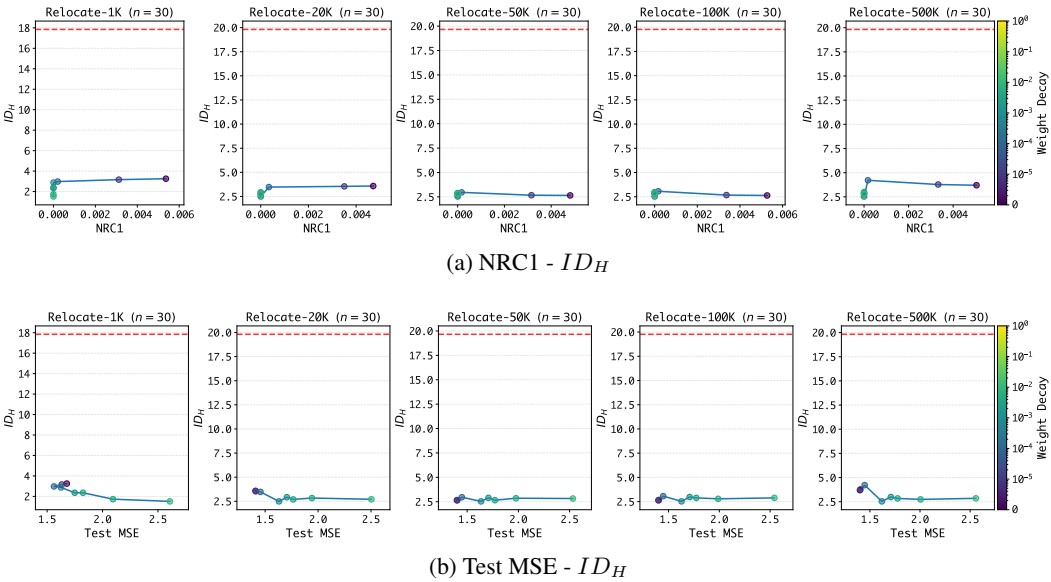

Figure 18: Relationship between $ID_H$ and NRC1 and Test MSE, for Relocate manipulation task, when varying data size ($\in \{1K, 20K, 50K, 100K, 500K\}$). The red dashed line represents $ID_Y$.

# I  Mathematical Analysis of Regression Collapse

To provide a principled explanation for the empirical phenomena observed in Section 4, we present a theoretical characterization of regression collapse. We first describe how weight decay induces a dimensional reduction in the feature space and subsequently formalize why this necessitates a reconstruction error.

**Why Weight Decay Causes Collapse?**  We analyze the effect of regularization through the Unconstrained Feature Model (UFM) framework employed in prior work on neural regression collapse (Andriopoulos et al., 2024), where $\lambda_{\mathbf{W}}$ and $\lambda_{\mathbf{H}}$ denote the penalties applied to the weights and features, respectively. By Theorem I.6 established in Appendix I.2, in the limit as $(\lambda_{\mathbf{W}}, \lambda_{\mathbf{H}}) \to 0^+$, the learned features $\mathbf{H}_\lambda$ converge to the minimum-norm solution of the unregularized problem. This solution lies entirely within an $n$-dimensional subspace, effectively eliminating the $(d - n)$-dimensional null space component of the features. As weight decay increases and starts violating the conditions for affine congruency discussed in Remark I.7, the model enters the over-compressed regime where $ID_H < ID_Y$.

**Why Collapsed Models Fail to Generalize?**  The over-compression creates an unavoidable reconstruction error, which can be formalized by treating the final layer as a smooth map $g$ from the feature manifold to the target space.

**Theorem I.1** (Non-surjectivity of Over-compressed Maps). *Let $\mathcal{M}_H$ be a smooth $m$-dimensional feature manifold and $\mathcal{N}_Y$ be a smooth $n$-dimensional target manifold, with $m < n$. A smooth map $g : \mathcal{M}_H \to \mathcal{N}_Y$ cannot be surjective; specifically, the image $g(\mathcal{M}_H)$ has Lebesgue measure zero in $\mathcal{N}_Y$.*

As proven in Appendix I.1 via Sard's Theorem, the condition $m < n$ in the over-compressed regime implies that the set of all possible model predictions is a proper subset of the target manifold. Geometrically, there will always be points on the target manifold that lie outside the model's predictive reach, rendering perfect reconstruction mathematically impossible. This explains why over-compressed models consistently correlate with high test MSE, as seen in Figure 4.

## I.1  Why Collapsed Models Fail to Generalize?

The proof of Theorem follows directly from Sard's theorem.

**Theorem I.2.** *Let $\mathcal{M}$ be a smooth $m$-dimensional manifold and $\mathcal{N}$ be a smooth $n$-dimensional manifold, with $m < n$. A smooth map $g : \mathcal{M} \to \mathcal{N}$ cannot be surjective, i.e., $g(\mathcal{M}) \neq \mathcal{N}$.*

*Proof.* Let $g : \mathcal{M} \to \mathcal{N}$ be a smooth map where $\dim(\mathcal{M}) = m$ and $\dim(\mathcal{N}) = n$, under the condition $m < n$. Consider an arbitrary point $p \in \mathcal{M}$. The differential of the map at this point, $dg_p : T_p\mathcal{M} \to T_{g(p)}\mathcal{N}$, is a linear transformation from the $m$-dimensional tangent space of $\mathcal{M}$ at $p$ to the $n$-dimensional tangent space of $\mathcal{N}$ at $g(p)$.

By the rank-nullity theorem, the rank of $dg_p$ is bounded by the dimension of its domain, so it holds that $\text{rank}(dg_p) \leq m$. Given that $m < n$, it follows that $\text{rank}(dg_p) < n$. A linear map is surjective if and only if its rank equals the dimension of its codomain; thus, $dg_p$ is not surjective.

As the choice of $p$ was arbitrary, this holds for all $p \in \mathcal{M}$. By definition, a point is critical if its differential is not surjective. Therefore, every point in the domain $\mathcal{M}$ is a critical point of $g$. The image of the set of critical points is the set of critical values. In this case, the set of critical values is the entire image of the map, $g(\mathcal{M})$.

By Sard's Theorem, the set of critical values of a smooth map has Lebesgue measure zero in the codomain. It follows that the image $g(\mathcal{M})$ has measure zero in $\mathcal{N}$. However, a smooth $n$-dimensional manifold (for $n \geq 1$) has positive Lebesgue measure. Since a set of measure zero cannot be equal to a set of positive measure, it must be that $g(\mathcal{M}) \neq \mathcal{N}$.

Therefore, the map $g$ is not surjective. □

This theorem provides the geometric foundation for understanding the failure of collapsed models. In our regression context, the learned features $\{\mathbf{h}_\theta(\mathbf{x})\}$ form a feature manifold $\mathcal{M}_H$ of dimension

$m = ID_H$, while the targets $\{\mathbf{y}\}$ lie on a target manifold $\mathcal{N}_Y$ of dimension $n = ID_Y$. The final layer of the network constitutes a smooth map from the feature manifold to the target space.

When a model is in the over-compressed regime ($ID_H < ID_Y$), the theorem's condition ($m < n$) is met. The direct consequence is that this smooth map cannot be surjective. This means the image of the feature manifold—the set of all possible predictions the model can generate—is a proper subset of the target manifold. Geometrically, there will always be points on the target manifold that lie outside the model's predictive reach. A perfect reconstruction is therefore impossible, as the model is fundamentally incapable of generating the full range of target data, leading to an unavoidable error.

## I.2 WHY WEIGHT DECAY LEADS TO COLLAPSE: ANALYSIS VIA THE UNCONSTRAINED FEATURE MODEL

In this section, we provide a theoretical explanation for why weight decay causes neural regression collapse through the lens of the Unconstrained Feature Model (UFM). The UFM is a simplified mathematical abstraction that does not capture all aspects of practical neural networks, but provides insight into the geometric mechanisms by which regularization constrains learned representations.

The UFM abstracts the neural regression problem by treating the feature extractor $h_\theta(\cdot)$ as producing arbitrary feature vectors $\mathbf{h}_i \in \mathbb{R}^d$ for each input $\mathbf{x}_i$, collected into a feature matrix $H \in \mathbb{R}^{d \times M}$. The final prediction is obtained via a linear map $W \in \mathbb{R}^{n \times d}$ and bias $\mathbf{b} \in \mathbb{R}^n$, giving predictions $\hat{Y} = WH + \mathbf{b}\mathbf{1}^\top$. The UFM objective is:

$$\min_{W,H,\mathbf{b}} \frac{1}{2M}\|WH + \mathbf{b}\mathbf{1}^\top - Y\|_F^2 + \frac{\lambda_W}{2}\|W\|_F^2 + \frac{\lambda_H}{2}\|H\|_F^2 \tag{1}$$

The model is "unconstrained" because $H$ can be any matrix in $\mathbb{R}^{d \times M}$, unlike in actual neural networks, where $H$ is constrained by the input data and network architecture. In the UFM, we typically have $d \gg n$, mirroring the overparameterized regime where feature dimension greatly exceeds target dimension.

The optimal bias is $\mathbf{b}^* = \bar{\mathbf{y}}$ where $\bar{\mathbf{y}} = \frac{1}{M}\sum_{i=1}^M \mathbf{y}_i$. Defining the centered target matrix $\tilde{Y} = Y - \bar{\mathbf{y}}\mathbf{1}^\top \in \mathbb{R}^{n \times M}$, the problem reduces to:

$$\min_{W,H} \frac{1}{2M}\|WH - \tilde{Y}\|_F^2 + \frac{\lambda_W}{2}\|W\|_F^2 + \frac{\lambda_H}{2}\|H\|_F^2 \tag{2}$$

We denote the empirical covariance by $\Sigma = \frac{1}{M}\tilde{Y}\tilde{Y}^\top \in \mathbb{R}^{n \times n}$, and assume $\Sigma$ is full rank with eigenvalues $\lambda_1 \geq \lambda_2 \geq \cdots \geq \lambda_n > 0$. A key quantity is $c = \lambda_W \lambda_H$.

We first restate the characterization of global minimizers when regularization is present, adapted from Theorem 4.1 of Andriopoulos et al. (2024). We focus on the regime where $c < \lambda_n$ since we are interested in the limiting behavior when $\lambda_H \to 0$ and $\lambda_W \to 0$.

**Theorem I.3** (Regularized UFM Solution, adapted from Andriopoulos et al. (2024)). *Suppose $0 < c < \lambda_n$. Define $A = \Sigma^{1/2} - \sqrt{c}I_n$. Then any global minimizer $(W_\lambda, H_\lambda)$ of equation 2 can be expressed as:*

$$W_\lambda = \left(\frac{\lambda_H}{\lambda_W}\right)^{1/4} A^{1/2} R \tag{3}$$

$$H_\lambda = \left(\frac{\lambda_W}{\lambda_H}\right)^{1/4} R^\top A^{1/2}(\Sigma^{1/2})^{-1}\tilde{Y} \tag{4}$$

*where $R \in \mathbb{R}^{n \times d}$ is any matrix satisfying $RR^\top = I_n$.*

*Proof.* This follows from Theorem 4.1 of Andriopoulos et al. (2024) with $c < \lambda_n$ implying $j^* = n$. $\square$

The regularized solution has a specific dimensional structure. Since $A^{1/2}(\Sigma^{1/2})^{-1}\tilde{Y} \in \mathbb{R}^{n \times M}$ and $R^\top \in \mathbb{R}^{d \times n}$, the matrix $H_\lambda \in \mathbb{R}^{d \times M}$ has columns lying in the column space of $R^\top$, which is at most

$n$-dimensional. Even though $H_\lambda$ lives in a $d$-dimensional ambient space with $d \gg n$, its columns are confined to an $n$-dimensional subspace.

When regularization is absent, the problem has infinitely many global minimizers, characterized by the following theorem from Andriopoulos et al. (2024).

**Theorem I.4** (Unregularized UFM Solutions, from Andriopoulos et al. (2024)). *When $\lambda_W = \lambda_H = 0$, a pair $(W, H)$ is a global minimizer of equation 2 if and only if $WH = \tilde{Y}$ and $W$ has full row rank. For any such $W$, the corresponding global minimizers in $H$ are:*

$$H_{unreg} = W^\dagger \tilde{Y} + (I_d - W^\dagger W)Z \tag{5}$$

*where $W^\dagger$ is the Moore–Penrose pseudoinverse and $Z \in \mathbb{R}^{d \times M}$ is arbitrary.*

*Proof.* See Theorem 4.3 of Andriopoulos et al. (2024). □

The structure in equation 5 decomposes solutions into two orthogonal components: $W^\dagger \tilde{Y}$ lies in the row space of $W$ (dimension at most $n$), while $(I_d - W^\dagger W)Z$ lies in the null space of $W$ (dimension exactly $d - n$). The Frobenius norm satisfies $\|H_{unreg}\|_F^2 = \|W^\dagger \tilde{Y}\|_F^2 + \|(I_d - W^\dagger W)Z\|_F^2$ by orthogonality. The minimum-norm solution is achieved when $Z = 0$, eliminating the $(d-n)$-dimensional null space component. When $Z \neq 0$, the null space allows $H$ to span up to $d$ dimensions, whereas $Z = 0$ confines $H$ to at most $n$ dimensions.

We now investigate what happens to the regularized solution as weight decay vanishes, examining the limit $\lambda_W, \lambda_H \to 0$ while maintaining $\lambda_H / \lambda_W = k > 0$.

**Lemma I.5** (Limiting Reconstruction). *Suppose $\lim_{\lambda_H \to 0, \lambda_W \to 0}(\lambda_H / \lambda_W) = k > 0$ and let $(W_\lambda, H_\lambda)$ be as in Theorem I.3. Then:*

$$\lim_{\lambda_W, \lambda_H \to 0} W_\lambda H_\lambda = \tilde{Y} \tag{6}$$

*Proof.* Since $c = \lambda_W \lambda_H \to 0$, Theorem I.3 applies for sufficiently small $\lambda_W, \lambda_H$. Multiplying:

$$W_\lambda H_\lambda = \left(\frac{\lambda_H}{\lambda_W}\right)^{1/4} A^{1/2} R \cdot \left(\frac{\lambda_W}{\lambda_H}\right)^{1/4} R^\top A^{1/2}(\Sigma^{1/2})^{-1}\tilde{Y} \tag{7}$$

$$= A^{1/2}(RR^\top)A^{1/2}(\Sigma^{1/2})^{-1}\tilde{Y} = A(\Sigma^{1/2})^{-1}\tilde{Y} \tag{8}$$

Substituting $A = \Sigma^{1/2} - \sqrt{c}I_n$:

$$W_\lambda H_\lambda = (\Sigma^{1/2} - \sqrt{c}I_n)(\Sigma^{1/2})^{-1}\tilde{Y} = (I_n - \sqrt{c}\Sigma^{-1/2})\tilde{Y} \tag{9}$$

As $c \to 0$, this yields $\tilde{Y}$. □

**Theorem I.6** (Limiting Solution Structure). *Under the assumptions of Lemma I.5, define:*

$$W_0 = \lim_{\lambda_W, \lambda_H \to 0} W_\lambda = k^{1/4}\Sigma^{1/4}R \tag{10}$$

$$H_0 = \lim_{\lambda_W, \lambda_H \to 0} H_\lambda = k^{-1/4}R^\top \Sigma^{-1/4}\tilde{Y} \tag{11}$$

*Then $(W_0, H_0)$ is a global minimizer of the unregularized problem with:*

$$H_0 = W_0^\dagger \tilde{Y} \tag{12}$$

*In particular, $H_0$ has no null space component (i.e., $Z = 0$ in Theorem I.4).*

*Proof.* From Lemma I.5, $W_0 H_0 = \tilde{Y}$, so $(W_0, H_0)$ is a global minimizer. Since $W_0 = k^{1/4}\Sigma^{1/4}R$ has full row rank, its pseudoinverse is:

$$W_0^\dagger = W_0^\top(W_0 W_0^\top)^{-1} = (k^{1/4}\Sigma^{1/4}R)^\top(k^{1/2}\Sigma^{1/2})^{-1} = k^{-1/4}R^\top \Sigma^{-1/4} \tag{13}$$

Thus $W_0^\dagger \tilde{Y} = k^{-1/4}R^\top \Sigma^{-1/4}\tilde{Y} = H_0$, confirming $Z = 0$. For any $H$ satisfying $W_0 H = \tilde{Y}$, we have $H = W_0^\dagger \tilde{Y} + (I_d - W_0^\dagger W_0)Z$ with:

$$\|H\|_F^2 = \|W_0^\dagger \tilde{Y}\|_F^2 + \|(I_d - W_0^\dagger W_0)Z\|_F^2 \geq \|H_0\|_F^2 \tag{14}$$

by orthogonality, with equality if and only if $Z = 0$. Thus $H_0$ is the minimum-norm solution. □

These results explain why weight decay causes dimensional collapse. Without regularization, $H$ can utilize the full $d$-dimensional space through the arbitrary null space component $(I_d - W^\dagger W)Z$, where the null space has dimension $d - n$. With any positive regularization, Theorem I.3 shows $H_\lambda$ is confined to an at most $n$-dimensional subspace. Theorem I.6 proves that even as $\lambda_W, \lambda_H \to 0^+$, the limiting solution has $Z = 0$, eliminating the $(d-n)$-dimensional null space component. This demonstrates that even infinitesimally small weight decay induces dimensional collapse by biasing toward the minimum-norm solution of the unregularized problem, which lies entirely in the $n$-dimensional row space of $W$. Since typically $d \gg n$, this represents a massive dimensional reduction from the ambient feature space to a low-dimensional subspace determined by the target structure.

*Remark* I.7 (Breakdown of Affine Congruence in the Over-Compressed Regime.). Theorem I.3, adapted from Theorem 4.1 of Andriopoulos et al. (2024), characterizes global minimizers of the regularized UFM objective in terms of the parameter

$$c := \lambda_W \lambda_H$$

and the eigenvalues $\lambda_1 \geq \cdots \geq \lambda_n > 0$ of the target covariance $\Sigma$. When $0 < c < \lambda_n$, the theorem yields $j^* = n$, and the solution satisfies

$$H_\lambda = \left(\frac{\lambda_W}{\lambda_H}\right)^{1/4} R^\top A^{1/2} (\Sigma^{1/2})^{-1} \tilde{Y}, \quad A = \Sigma^{1/2} - \sqrt{c}\, I_n,$$

with $A \succ 0$. In this regime, $A^{1/2}$ is full rank and the map relating $H_\lambda$ and $\tilde{Y}$ is an invertible affine transformation on the support of $\tilde{Y}$. Consequently, the learned features and targets are affine congruent, implying preservation of intrinsic dimension:

$$ID_H = ID_Y.$$

In contrast, when $c \geq \lambda_n$, Theorem 4.1 of Andriopoulos et al. (2024) yields $j^* < n$, and the solution involves truncated matrices $[A^{1/2}]_{j^*}$ and $[\Sigma^{1/2}]_{j^*}$ obtained by retaining only the eigenspaces corresponding to eigenvalues $\lambda_i > c$. In this case, the resulting feature matrix satisfies

$$\mathrm{rank}(H_\lambda) \leq j^* < n,$$

and the affine map relating $H_\lambda$ and $\tilde{Y}$ is no longer invertible on the support of $\tilde{Y}$. As a consequence, affine congruency is not guaranteed to hold. This regime allows for a reduction in intrinsic dimension,

$$ID_H < ID_Y,$$

which is precisely what we observe empirically in the over-compressed region. This observation highlights the diagnostic value of intrinsic dimension: deviations between $ID_H$ and $ID_Y$ signal departure from the regime in which the assumptions underlying affine congruency hold. Although the parameter $c$ is not directly observable in practice, as it arises from the UFM abstraction rather than explicit training dynamics, intrinsic dimension provides an empirical proxy for this regime. In particular, the transition from $ID_H \approx ID_Y$ to $ID_H < ID_Y$ indicates that the effective regularization has crossed the threshold $c = \lambda_n$, placing the model in the over-compressed regime and violating the conditions under which affine congruency is expected.

## J    LIMITATIONS AND FUTURE WORK

While our work provides new geometric insights into neural multivariate regression through intrinsic dimension analysis, some limitations remain. Although we provide some theoretical results explaining why weight decay causes collapse (Appendix I.2) and why collapsed models often fail (Appendix I.1), a complete theoretical characterization of the relationship between intrinsic dimension and generalization is not yet available. Additionally, the 2-NN estimator we employ provides reliable estimates for intrinsic dimensions below approximately 20. For extremely high-dimensional target spaces or feature representations, alternative estimation methods may be necessary. Finally, our practical guidelines rely on adjusting standard hyperparameters (weight decay, model depth, dropout) to indirectly control $ID_H$. A more principled approach would be to optimize $ID_H$ during training; however, the 2-NN estimator is non-differentiable and cannot be optimized via backpropagation.

Several promising directions emerge from our findings. Deriving generalization bounds that incorporate intrinsic dimension would provide a rigorous theoretical foundation for the empirical relationships we observe. Theoretical frameworks beyond the UFM could offer additional perspectives on how network architecture, training dynamics, and regularization jointly determine the intrinsic dimension of learned representations. Differentiable intrinsic dimension estimation remains largely underexplored; developing robust, efficient, and scalable differentiable estimators that enable direct control of $ID_H$ during training via backpropagation represents an important research direction. Such advances would allow practitioners to explicitly target desired intrinsic dimensions rather than adjusting hyperparameters indirectly, and would broaden the applicability of our analysis to higher-dimensional settings. Finally, exploring whether intrinsic dimension provides similar insights for other learning paradigms, such as generative modeling, reinforcement learning, and multi-task learning, could offer a unifying geometric perspective across domains.

# K  SCIENCE OF DL IMPROVEMENT CHALLENGE SUBMISSION

## K.1  WHAT MODEL ARE YOU TARGETING?

*Provide a summary of the problem the deep net model is designed to solve. Good summaries should outline the state of the literature, provide an overview that domain experts would consider reasonable, and cite relevant sources.*

We study standard model architectures, including multi-layer perceptron (MLP) policies operating on raw states for state-based control tasks; and convolutional neural network (CNN) feature encoders followed by an MLP policy head, for vision-based tasks. Despite their simplicity, these architectures remain the mainstream choice in classic online and offline reinforcement learning (RL) algorithms(Fujimoto et al., 2018; Haarnoja et al., 2018; Emmons et al., 2022; Yarats et al., 2022; Kostrikov et al., 2022; Tarasov et al., 2023; Xu et al., 2024), as naively scaling model size often leads to instability or degraded performance(Ota et al., 2021; Nauman et al., 2024a; Lyle et al., 2024; Ceron et al., 2024b), and more advanced model architectures are still ongoing research questions with less understanding (Nauman et al., 2024b; Lee et al., 2025a;b; Wang et al., 2025).

Recent literature highlights the importance of regularization—such as weight decay(Nauman et al., 2024b), dropout(Hiraoka et al., 2022), and layer normalization(Nauman et al., 2024b; Wang et al., 2025)—to stabilize training, improve generalization, and enable limited scaling in RL settings (Liu et al., 2021; Kumar et al., 2022; Li et al., 2023b). The regularized models studied in this work align with these practices, making our findings potentially applicable to these commonly used RL pipelines.

## K.2  HOW DO YOUR RESULTS CONTRIBUTE—OR COULD POTENTIALLY CONTRIBUTE—TO UNDERSTANDING THESE MODELS?

*What aspects of the models become better understood thanks to your work?*

Visual control is particularly interesting and challenging because each pixel frame provides only partial information about the system state, forming a Partially Observed Markov Decision Process (POMDP) (Yarats et al., 2022). Appendix H.1 experiments with such a vision control task by behavior-cloning an expert demonstration dataset generated from the simulated Cheetah_run environment. We use a 4-layer CNN image encoder, followed by a 3-layer MLP policy to produce actions.

Our empirical results have direct implications for evaluating real-world control models, where evaluation is often expensive, unsafe, and risky (Levine et al., 2020). Validation MSE provides little indication of actual control performance in the real tasks. Practitioners must therefore rely on costly real-environment interactions to tune hyperparameters and assess the policy. Figure 19 shows that the relationship between $ID_H$ and $ID_Y$ not only predicts test MSE but also aligns with the *true* control performance obtained from environment interaction. In particular, the relationship between $ID_H$ and normalized model performance exhibits the opposite 'U-shape' behavior, achieving its best score when $ID_H \approx ID_Y$, mirroring the sweet-spot identified in Section 4.

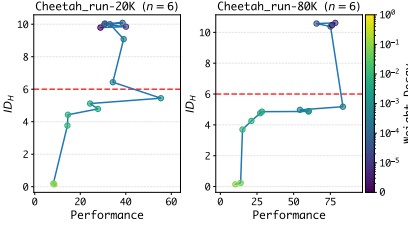

Figure 19: Relationship between $ID_H$ and normalized model performance ($\in [0, 100]$; the higher, the better). The horizontal red dashed line represents $ID_Y$.

## K.3  HOW DO YOU EXPECT YOUR SUBMISSION TO INFLUENCE FUTURE WORK?

*Propose ways in which your insights, findings, or methodologies could shape subsequent research directions, model design choices, or scientific applications.*

Prior work often associates strong online/offline RL performance with high feature rank of critics and actors (Kumar et al., 2021; Lyle et al., 2022; Nauman et al., 2024a). Our findings suggest an alternative perspective: improved performance may arise from constraining representation geometry such that feature intrinsic dimension matches task-relevant targets, rather than maximizing expressivity. This insight motivates future work on geometry-aware representation learning for RL, potentially grounded in the manifold hypothesis, as done recently by Li & He (2025) and Tiwari et al. (2025).

