# OpenReview forum: "Geometric Properties of Neural Multivariate Regression: An Empirical Study"
_ICLR.cc/2026/Workshop/Sci4DL — Sci4DL 2026_

### Official Review · Reviewer_YY3d · 2026-02-14

**Fit:** 3
**Significance:** 3
**Confidence:** 2

**Summary:**

The paper articulates a clear open question—*Why does neural collapse
hinder generalization in multivariate regression, in contrast to its beneficial role in classification*—and motivates intrinsic dimension as a more refined geometric lens than PCA-based collapse metrics. It formalizes collapse using an NRC1 metric based on PCA reconstruction residuals, and introduces intrinsic dimension (ID) estimated by a 2-NN method to capture nonlinear (low-dimension) geometry beyond PCA. The empirical narrative linking generalization to the $ID_H–ID_Y$ relationship (including an over-compressed regime $ID_H < ID_Y $ and task-dependent behavior when $ID_H ≥ ID_Y$) is stated explicitly and tied to concrete figure-based observations. Moreover, the observations support that ID is more sensitive than the PCA-based NRC1 metric. The paper is strengthened by detailed experimental configurations (architectures/regularization ranges...) and further investigations on other perspectives (e.g. the evolution of intrinsic dimension during training...). The experiments in the appendix are also detailed and well-designed, with related theorems explaining these observations.

**Strengths:**

**Significance of the raised question**: Good, Interesting, and Important.

**Evidence supporting the viewpoint answering the raised question**: (Almost) Fully convincing.

**Soundness of the methodology**: Good.

**Strengths:**
1. The experimental results validate the fact that *the intrinsic dimension(ID) is more refined than NRC1 metric, especially in over-compressed settings where $ID_H < ID_Y$*, as shown in Figure 3. (As shown in Figure 2, it's intuitive that ID is closer to the essence of the distribution than NRC1.)

2. A large number of experiments are done to support the paper's viewpoint. And there are also experiments regarding other related perspectives(e.g., the relation between regularization and ID), which are necessary to further understand the formulation of the phenomenon.

3. Appendix I.1 provides a concrete geometric statement—non-surjectivity (*Even zero-measure of image*) when (smooth) mapping from an m-dimensional feature manifold to an n-dimensional target manifold with m<n—directly linking $ID_H<ID_Y$ to unavoidable approximation error

4. Experimental details are provided in the Appendix.

5. The paper is well-written and easy to read.

**Suggestions:**

**Major Concerns and Suggestions**:

1. If I understand it right, the 2-NN method for estimating ID relies on the local Poisson process assumption. So how can we guarantee that *the assumption, or other assumptions, are satisfied by the distribution of layer-output and target manifold?* The experimental results in the paper are fully convincing to me, but I'm still confused about why the estimation of ID is (approximately) correct in the layer-output and target manifold.

2. Assume the estimation is (approximately) correct, then in situations where $ID_H < ID_Y$, the image under the (smooth) output layer of the last-layer feature (corresponds to $ID_P$)  is of zero measure in the target manifold. So the training error might be large, as shown in your experiment. In this situation, generalization is meaningless. I wonder what would happen when we use a more powerful output layer to get a lower training error, though it may be overfit.

**Minor:**
1. In Appendix C, *"a sphere in 10D has d=2"*: better to say it's a sphere in $\mathbb{R}^3$ embedding in 10D.

2. I think it's better to visualize the manifold of the last-layer feature, the prediction, and the target vector in some simple tasks.

3. In your training process, how much time does the estimation of ID cost? In large-scale settings, is 2-NN an efficient way to measure ID?

4. Add additional noise models or real-world multivariate regression datasets where target noise is naturally present, to test whether the U-shaped behavior persists beyond the projection/domain-mismatch construction.

5. Are there any indicator in the training process that indicate $ID_H \approx ID_Y$? If so, I think it's better to write it down, as it can help save computational cost while maintain good generalization.

---

### Official Review · Reviewer_o1Js · 2026-02-24

**Fit:** 3
**Significance:** 3
**Confidence:** 2

**Summary:**

This paper shows neural collapse for multivariate regression is usually harmful to generalization, in contrast with neural collapse for classification being beneficial. Metrics for neural regression collapse are defined, with strong correlation between test error and derived NRC1/ID metrics for robotic control tasks. Intuitive explanations are given for why neural collapse harms generalization for regression tasks.

**Strengths:**

An intuitive description is given for why neural collapse is detrimental to having neural networks perform multivariate regression, an incredibly common task, with the metrics used being applicable even to complex domains such as robotic control. The presentation was clear and easy to follow. Defined metrics for intrinsic dimension and NRC1 are qualitatively consistent, giving strong evidence that the underlying phenomenon is being properly described. The question answered by this paper appears to be fundamental to the field of deep learning, at least for regression settings.

**Suggestions:**

The only main recommendation I have is to explore more small-scale, limited problems where it's easier to draw direct causal relations between NRC1/ID and generalization. It's possible, though seemingly unlikely, that the results are purely correlational and dependent on hidden variables with a separate, causational explanation being difficult to spot. Without precise control over an incredibly simple toy problem, it's impossible to rule out that possibility.

This was likely not included in the main body for space constraints, but some discussion of the difference between NC in classification and regression should be included. Are the NRC1 and ID metrics useful when looking at NC in classifying networks? Why or why not?

Additionally one small point, in Figure 4, it seems the training curves were included in the (d, 2) and (e) plots, along with there being two different (d) labels.

---

### Meta-Review · Area_Chair_Npgd · 2026-02-28

**Recommendation:** Accept

**Metareview:**

Intuitive conclusion, and reviewers like it. Nice!

---

### Decision · Program_Chairs · 2026-03-02

Accept